# The transpeptidase PBP2 governs initial localization and activity of the major cell-wall synthesis machinery in *E. coli*

Gizem Özbaykal[1,2†], Eva Wollrab[1†], Francois Simon[1], Antoine Vigouroux[1,3,4], Baptiste Cordier[1], Andrey Aristov[1], Thibault Chaze[5], Mariette Matondo[5], Sven van Teeffelen[1]*

[1]Microbial Morphogenesis and Growth Lab, Institut Pasteur, Paris, France; [2]Université Paris Diderot, Sorbonne-Paris-Cité, Paris, France; [3]Synthetic Biology Lab, Institut Pasteur, Paris, France; [4]Université Paris Descartes, Sorbonne-Paris-Cité, Paris, France; [5]Proteomics Platform, Institut Pasteur, Paris, France

**Abstract** Bacterial shape is physically determined by the peptidoglycan cell wall. The cell-wall-synthesis machinery responsible for rod shape in *Escherichia coli* is the processive 'Rod complex'. Previously, cytoplasmic MreB filaments were thought to govern formation and localization of Rod complexes based on local cell-envelope curvature. Using single-particle tracking of the transpeptidase and Rod-complex component PBP2, we found that PBP2 binds to a substrate different from MreB. Depletion and localization experiments of other putative Rod-complex components provide evidence that none of those provide the sole rate-limiting substrate for PBP2 binding. Consistently, we found only weak correlations between MreB and envelope curvature in the cylindrical part of cells. Residual correlations do not require curvature-based Rod-complex initiation but can be attributed to persistent rotational motion. We therefore speculate that the local cell-wall architecture provides the cue for Rod-complex initiation, either through direct binding by PBP2 or through an unknown intermediate.

**\*For correspondence:**
sven.vanteeffelen@gmail.com

[†]These authors contributed equally to this work

**Competing interests:** The authors declare that no competing interests exist.

## Introduction

The peptidoglycan (PG) cell wall is the major load-bearing structure of the bacterial cell envelope and physically responsible for cell shape (*Vollmer et al., 2008*). Rod-like cell shape in *Escherichia coli* requires peptidoglycan synthesis by stable multi-enzyme 'Rod complexes' containing the trans-glycosylase RodA, the transpeptidase PBP2, the transmembrane protein RodZ, and the actin homo-log MreB (*Cho et al., 2016*; *Emami et al., 2017*; *Meeske et al., 2016*; *Morgenstein et al., 2015*; *Typas et al., 2012*). All of these proteins move persistently around the cell circumference at similar speeds (*Cho et al., 2016*; *Morgenstein et al., 2015*; *van Teeffelen et al., 2011*), suggesting that these proteins stably associate for processive cell-wall insertion. Colocalization of MreB and RodZ (*Alyahya et al., 2009*; *Bendezú et al., 2009*; *Morgenstein et al., 2015*) supports this idea.

Other proteins (MreC, MreD, PBP1a, and PBP1b) are possibly also part of these complexes (*Banzhaf et al., 2012*; *Cho et al., 2016*; *Contreras-Martel et al., 2017*; *Kruse et al., 2004*; *Morgenstein et al., 2015*). MreC activates PBP2 (*Contreras-Martel et al., 2017*; *Rohs et al., 2018*). However, the shape defect of a *mreCD* deletion is partially suppressed by a hyperactive PBP2 point mutant (*Rohs et al., 2018*), suggesting that neither MreC nor MreD are strictly necessary for Rod-complex assembly or function. The bi-functional class-A penicillin-binding proteins PBP1a and PBP1b interact with PBP2 and RodZ, respectively (*Banzhaf et al., 2012*; *Morgenstein et al., 2015*), and PBP2 activates PBP1a glycosyltransferase activity in vitro (*Banzhaf et al., 2012*). However, Rod-complex rotational motion is independent of class-A PBP activity (*Cho et al., 2016*). Furthermore,

single-molecule tracking suggests that any possible association of PBP1a or PBP1b with the Rod complex is short lived (*Cho et al., 2016*). Similar to *mreCD*, a *rodZ* deletion can also be suppressed by point mutations in PBP2, RodA, or MreB (*Shiomi et al., 2008*). Summarizing, it emerges, that RodA, PBP2, and MreB form the core of the Rod complex (*Rohs et al., 2018*). On the contrary, the determinants of Rod-complex spatial distribution and activity, which are ultimately responsible for cell shape, remain less well understood.

MreB filaments are intrinsically curved (*Hussain et al., 2018*; *Salje et al., 2011*). This curvature likely stabilizes their circumferential orientation (*Billaudeau et al., 2019*; *Hussain et al., 2018*; *Olshausen et al., 2013*; *Ouzounov et al., 2016*; *Wang and Wingreen, 2013*) and the circumferential orientation of Rod complex motion (*Errington, 2015*; *Hussain et al., 2018*).

Previously, it has been suggested that MreB filaments provide a platform that recruits other Rod-complex components to the site of future cell-wall synthesis (*Errington, 2015*; *Shi et al., 2018*; *Surovtsev and Jacobs-Wagner, 2018*). Accordingly, MreB filaments might be responsible for the initial localization of Rod complexes. Ursell et al. and others suggested that MreB filaments are attracted to sites of specific two-dimensional cell-envelope curvature (*Billings et al., 2014*; *Shi et al., 2018*; *Ursell et al., 2014*) based on mechanical properties of MreB filaments and RodZ-MreB interactions (*Bratton et al., 2018*; *Colavin et al., 2018*). However, correlations could also come about indirectly, for example through a curvature-independent depletion of MreB from highly curved cell poles (*Kawazura et al., 2017*) or through persistent motion (*Hussain et al., 2018*; *Wong et al., 2017*; *Wong et al., 2019*). Therefore, the initial localization of Rod complexes could in principle be governed by factors different from MreB. We thus wondered, whether the cell wall itself could provide a local cue for the initiation of Rod complexes, independently of cell-envelope curvature. Such a local cue would have to be sensed by a protein with a periplasmic domain that can possibly bind the cell wall.

An obvious candidate is the transpeptidase PBP2. For its cross-linking activity PBP2 must bring together donor peptides on nascent glycan strands and acceptor peptides in the cell wall. Binding of PBP2 to the existing cell wall could therefore provide an alternative mechanism of Rod-complex initiation. In support of this hypothesis, a PBP2(L61R) mutant shows increased cell-wall synthetic activity and affects the distribution of MreB-actin filament length (*Rohs et al., 2018*). Further support comes from localization studies in *E. coli* (*Lee et al., 2014*) and *Caulobacter crescentus* (*Dye et al., 2005*; *Hocking et al., 2012*). The spatial distributions of PBP2 and MreB only partially overlap, which is compatible with the hypothesis that PBP2 finds sites for activity independently of MreB.

Here, we used single-molecule tracking to demonstrate that diffusive PBP2 molecules stably bind to an immobile component in the cell envelope. Bound molecules transition between immobile and persistently moving states, the latter depending on Rod-complex activity. Interestingly, MreB filaments are not the binding substrate for PBP2, nor does MreB determine locations of PBP2 binding. Depletion and localization experiments of other known Rod-complex components suggest that none of those provide the sole substrate for PBP2 binding. These observations suggest that PBP2 or an unknown protein determine the initial localization of newly forming Rod complexes. We speculate that the cell wall itself provides the cue for Rod-complex initiation. In support of our observations, we found that MreB filaments are likely not recruited to regions of particular cell-envelope curvature in filamentous and normal cells, contrary to *Ursell et al. (2014)*. Specifically, we found only weak MreB-curvature correlations once cell poles were excluded from the analysis. Residual correlations were attributed to weak spontaneous cell bending, suggesting that they are caused by persistent rotational motion (*Wong et al., 2017*). Finally, we also found that fast diffusing molecules cannot contribute to processive Rod-complex activity due to limitations of diffusion, contrary to *Lee et al. (2014)*.

## Results

### PBP2 enzymes can be quantitatively separated into diffusive and bound fractions

To study the role of PBP2 for the formation of Rod complexes we characterized its different states of motion, which are potentially representative of different states of substrate binding and activity. We imaged a functional, N-terminal protein fusion of the photo-activatable fluorescent protein

PAmCherry to PBP2 (*Lee et al., 2014*). The fusion is expressed from the native *mrdA* locus at a level similar to the wild-type protein according to quantitative mass spectrometry (*Figure 1—figure supplement 1D*; *Supplementary file 1a*) and Bocillin labeling experiments (*Figure 1—figure supplement 1A* and also *Lee et al., 2014*). The strain carrying the fusion maintains rod-like cell shape with only slight deviations of average cell diameter and length (*Figure 1—figure supplement 1B*) and does not show any growth defect (*Figure 1—figure supplement 1C*; *Lee et al., 2014*).

We obtained single-molecule tracks by single-particle tracking *PhotoActivatable Localization Microscopy* (sptPALM) (*Manley et al., 2008*) in total internal reflection fluorescence (TIRF) mode, which restricts the observation to the bottom part of the cell. We first imaged PBP2 molecules at high frequency (intervals of 60 ms). We found both spatially extended trajectories, corresponding to fast diffusing molecules, and trajectories that appeared as localized, corresponding to immobile or slowly moving molecules (*Figure 1A*, *Figure 1—video 1*).

We confirmed the presence of two distinct fractions of diffusing and localized molecules based on the distributions of jump lengths during different time lags (*Figure 1B*). Specifically, we fit those distributions to a two-state or a three-state diffusion model using the Spot-On tool (*Hansen et al., 2018*) (Materials and methods; *Supplementary file 2*). Both models contain as a special case an immobile population, and they lead to very similar results (*Figure 1B*; *Figure 1—figure supplement 2*). We found a diffusive population with average diffusion constant $D = 0.04$ μm$^2$/s containing 74–78% of all enzymes, and a second population with zero diffusion constant containing 22–26% of all molecules (*Supplementary file 1a*). The three-state model led to slightly better fits, predicting a larger fast-diffusing and a smaller slow-diffusing population with diffusion constants of $D_1 = 0.015 \pm 0.005$ μm$^2$/s and $D_2 = 0.055 \pm 0.006$ μm$^2$/s, respectively. Because the true diffusive behavior might be more complex, we report the average diffusion constant in the following. We confirmed our findings using an alternative method that is based on the distribution of single-track effective diffusion constants, yielding a slightly lower bound fraction (*Figure 1—figure supplement 3*).

Upon overexpression by about three- to six-fold (*Figure 1—figure supplement 1D*) using the vector pKC128 that expresses PAmCherry-PBP2 from the native *mrdAB* promoter (*Lee et al., 2014*) the fraction of molecules with near-zero diffusion constant decreased to $12.3 \pm 1.1\%$ (*Figure 1H–I* and *Figure 1—figure supplement 2*). Here, the three-state diffusion model gave significantly better fits (*Figure 1—figure supplement 2*). We therefore decided to use the three-state model throughout the rest of the paper. We also noticed that the average diffusion constant of the diffusive population decreased from $D = 0.042 \pm 0.004$ μm$^2$/s to $0.025 \pm 0.002$ μm$^2$/s. We will come back to this latter point further down.

While diffusing molecules are likely enzymatically inactive and possibly searching for new sites of cell-wall insertion (see further down), localized molecules are likely bound to an immobile substrate or part of slowly moving Rod complexes with anticipated speeds of 10–40 nm/s (*Cho et al., 2016*; *van Teeffelen et al., 2011*).

## Bound PBP2 molecules are either persistently moving or immobile

To test whether all or part of the bound molecules were moving persistently we imaged PBP2 molecules at low frequency, taking images with an exposure time of 1 s and intervals of 3.6 s. The long exposure time effectively smears out the fluorescence of fast diffusing molecules, allowing us to detect the positions of individual bound molecules. Using this protocol, we found molecules that moved persistently, were immobile, or showed transitions between these two states (*Figure 1C–E*, *Figure 1—video 2*, and *Figure 1—figure supplement 4*).

Persistently moving molecules showed similar distributions of speed and orientation as a functional msfGFP-MreB fusion (*Ouzounov et al., 2016*) expressed from the native *mreB* locus (*Figure 1F,G*; *Cho et al., 2016*), in agreement with previous measurements (*Cho et al., 2016*; *van Teeffelen et al., 2011*). For accurate velocity measurements, we obtained these results from movies acquired with a shorter time interval of 1 s. Straight tracks representing persistently moving molecules were selected based on the mean squared displacements (MSD) (*Figure 1—figure supplement 5*).

Because PBP2 molecules show transitions between different states in single trajectories, we quantified immobile and persistent states locally in time. Specifically, we classified motion states using a single threshold on the mean velocity during four consecutive time steps in movies acquired with 3.6

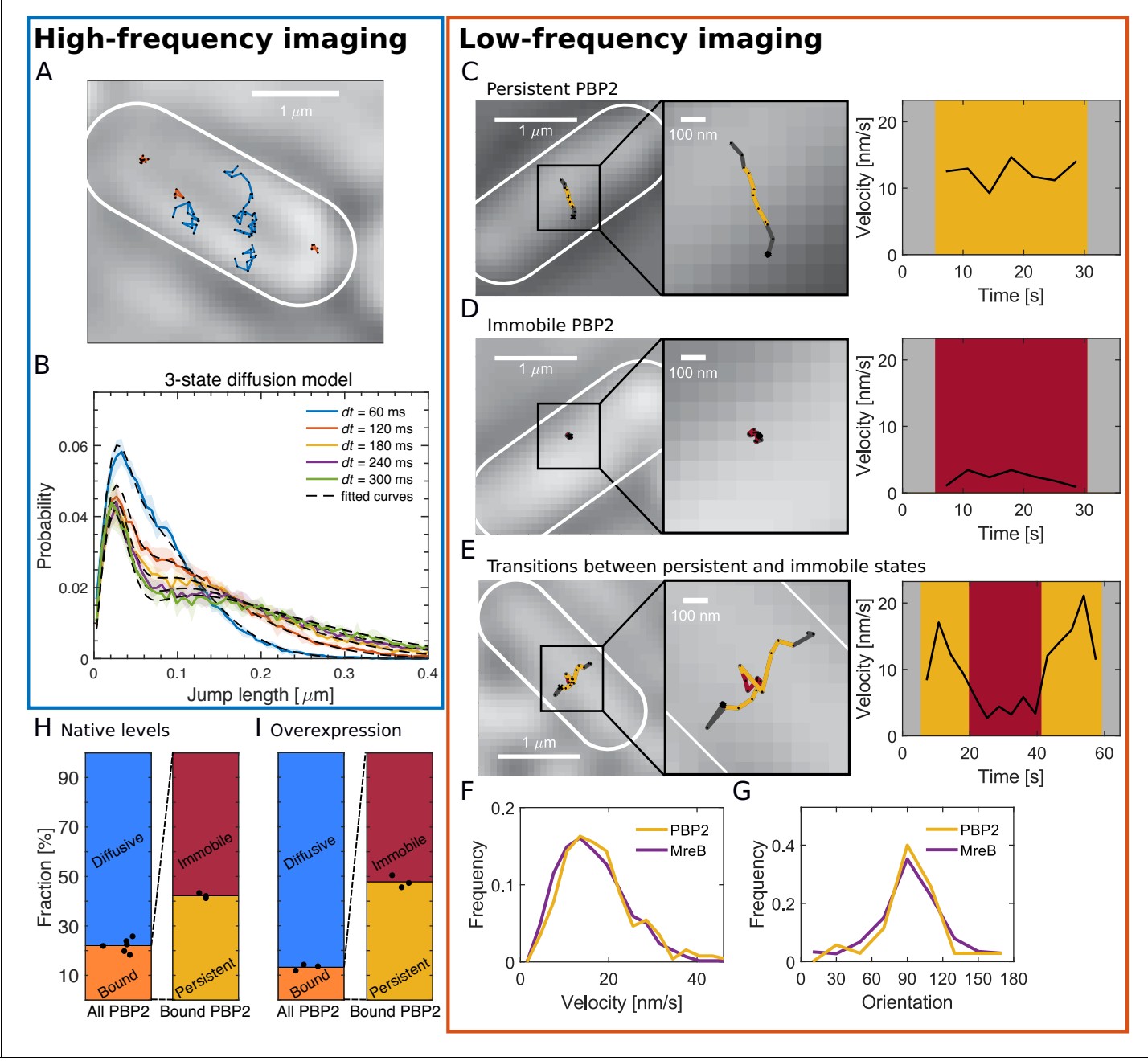

**Figure 1.** PBP2 molecules reside in diffusive, immobile, or persistently moving states. (**A**) Representative trajectories of PAmCherry-PBP2 molecules (TKL130) obtained by high-frequency imaging (time interval 60 ms) reveals diffusive (blue) and bound (orange) molecules. (**B**) Probability distribution of single-molecule jump lengths (solid lines, colored) and fit (dashed, black) using a three-state-diffusion model for different time intervals. 78% of PBP2 move diffusively with $<D>$ = 0.042 µm²/s while 22% are immobile. The shaded area indicates standard deviation between six biological replicates. (**C–E**) Low-frequency imaging (3.6 s with 1 s exposure time) reveals that bound PBP2 molecules are either persistently moving (**C**) or immobile (**D**), according to the instantaneous PBP2 velocity. PBP2 molecules show transitions between persistent and immobile states (**E**). (**F–G**) Persistently moving PBP2 and MreB filaments show similar speeds (**F**) and orientations of motion (orientation measured with respect to the cell centerline) (**G**). (**H–I**) Average fractions of bound, diffusive, persistently moving, and immobile PAmCherry-PBP2 at native levels (TKL130) (**H**) or if overexpressed (TKL130/pKC128) (**I**). Dots show biological replicates.

The online version of this article includes the following video, source data, and figure supplement(s) for figure 1:

**Source data 1.** Table containing all data presented in *Figure 1* and *Figure 1—figure supplements 1–10*.
**Figure supplement 1.** Comparison of PBP2-PAmCherry expressing cells and WT.
**Figure supplement 2.** Comparing 2- and 3-state-diffusion models to fit experimental data through Spot-On.

*Figure 1 continued on next page*

*Figure 1 continued*

**Figure supplement 3.** An alternative approach to fit a two-state diffusion model, based on the distribution of effective diffusion constants.
**Figure supplement 4.** Transitions between immobile and persistent states.
**Figure supplement 5.** Analysis of bound PAmCherry-PBP2 molecules.
**Figure supplement 6.** Quantitative analysis of persistent and immobile states based on computational simulations.
**Figure supplement 7.** Number of bound PBP2 molecules increases with increasing PBP2 levels.
**Figure supplement 8.** Comparison of AV127, msfGFP-PBP2 (TU230(attLHC943)), msfGFP-PBP2(L61R) (TU230(attLHC943)), and WT strains.
**Figure supplement 9.** Time-dependent effect of low msfGFP-PBP2 expression.
**Figure supplement 10.** Low-frequency tracking of msfGFP-PBP2 and msfGFP-PBP2(L61R) cells under different induction levels.
**Figure 1—video 1.** High-frequency imaging of PAmCherry-PBP2.
https://elifesciences.org/articles/50629#fig1video1
**Figure 1—video 2.** Low-frequency imaging of PAmCherry-PBP2.
https://elifesciences.org/articles/50629#fig1video2
**Figure 1—video 3.** High-frequency imaging of msfGFP-PBP2.
https://elifesciences.org/articles/50629#fig1video3
**Figure 1—video 4.** Low-frequency imaging of msfGFP-PBP2.
https://elifesciences.org/articles/50629#fig1video4

s interval (*Figure 1C–E*). Window size and velocity threshold (8 nm/s) were chosen based on computationally simulated tracks (*Figure 1—figure supplement 6A-C*). In confirmation of our two-state model, we found good agreement between the average MSD obtained from experiment and simulation for immobile and persistent segments, respectively (*Figure 1—figure supplement 6D-F*). Other states of motion are therefore likely not present. Using this criterion, we found a persistent fraction of 42.2 ± 1.1% of all bound molecules, while 57.8 ± 1.1% remained immobile (*Figure 1H*).

Upon overexpression of PAmCherry-PBP2 as above, we found that the persistent fraction remained nearly constant (*Figure 1I*; *Supplementary file 1a*), suggesting that the number of active Rod complexes, $N_{\mathrm{active}} = N_{\mathrm{PBP2}}bp$ (with $N_{\mathrm{PBP2}}$ the number of PBP2 molecules, and $b$ and $p$ the bound and persistent fractions, respectively), increased by about two-fold (*Figure 1—figure supplement 7B*. This finding suggests that PBP2 is an important limiting component for the number of active Rod complexes. This viewpoint is consistent with the recent report that a hyperactive PBP2 point mutant (L61R) increased the overall amount of active Rod complexes (*Rohs et al., 2018*). We will come back to this mutant in the next section.

## msfGFP-PBP2 fusion confirms findings and demonstrates increased PBP2 binding upon PBP2 depletion

We tested our findings using a strain that carries a functional msfGFP-PBP2 fusion (*Cho et al., 2016*) expressed at the native *mrdA* locus (AV127). Surprisingly, PBP2 levels were about four-fold lower than in the wildtype according to quantitative mass spectrometry. Yet, the strain showed normal cell shape and growth (*Figure 1—figure supplement 8A*). Tracking msfGFP-PBP2 at high frequency after initial pre-bleaching, we found a fraction of bound PBP2 molecules of 36.5 ± 5% (*Figure 1—figure supplement 8C*, *Figure 1—video 3*), which is higher than in the case of PAmCherry-PBP2.

To test whether the higher fraction was due to reduced PBP2 levels, we also used a strain that carries the same msfGFP-PBP2 fusion under IPTG-inducible control as the sole copy of PBP2 (MG1655 *mrdA::aph* (P$_{\mathrm{lac}}$::*msfgfp-mrdA*)) (*Cho et al., 2016*). Indeed, we found that the bound fraction decreased with increasing induction level (*Figure 1—figure supplement 8C*), qualitatively similar to the PAmCherry fusion.

To our surprise, near-wild-type levels observed at low induction (5 µM IPTG) led to loss of rod shape at long times (*Figure 1—figure supplement 9A*), even though average PBP2 levels were close to wild-type levels after 6 hr (*Figure 1—figure supplement 8B*) and remained constant during long-term exponential growth (*Figure 1—figure supplement 9A*). Since AV127 did not display shape defects despite lower average levels of msfGFP-PBP2, we reasoned that shape defects might be due to the noisier expression from the inducible promoter.

Long-term shape changes in the inducible strain were accompanied by an almost two-fold increase of the bound fraction (*Figure 1—figure supplement 9B*). We therefore speculated that

stronger PBP2 binding might be a response to the increased need for Rod-complex activity. We will come back to a potential feedback between PBP2 binding and cell-wall architecture further down.

As a test of our method, we also tracked an msfGFP fusion to the PBP2 point mutant (L61R) recently characterized by *Rohs et al. (2018)*. Based on slow-frequency imaging, they reported that the number of bound msfGFP-PBP2(L61R) was about two-fold increased as compared to msfGFP-PBP2 (*Rohs et al., 2018*). We confirmed this finding quantitatively, using the same inducible PBP2 fusions (*Rohs et al., 2018*; *Figure 1—figure supplement 8C*).

Among the bound wildtype molecules we found a fraction of 76–83% persistently moving molecules, depending on strain and expression level (*Figure 1—figure supplement 10A*, *Supplementary file 1a*, *Figure 1—video 4*), which is significantly higher than in the case of the PAmCherry fusion. We therefore reasoned that the PAmCherry fusion might have reduced functionality. However, similarly to the PAmCherry fusion, the persistent fraction remained nearly constant over the 20-fold range of expression levels (according to Western Blot) (*Figure 1—figure supplement 10A*, *Supplementary file 1a*). Therefore, PBP2 activity is predominantly limited by the number of bound PBP2 molecules, similar to the PAmCherry fusion.

We also measured persistent motion in the PBP2(L61R) mutant and found that the persistent fraction is reduced in comparison to the wild-type protein (*Figure 1—figure supplement 8A*, p-value 0.016).

In summary, these results are compatible with our findings with the PAmCherry fusion.

## PBP2 molecules show long persistent runs

The classification into different motion states at the sub-trajectory level allowed us to extract average transition rates between immobile and persistently moving states, $k_{ip}$ and $k_{pi}$, respectively (*Figure 2A–B*). Depending on the fluorescent-protein fusion, we found values of $k_{ip}$ between 0.014–0.063 s$^{-1}$, and of $k_{pi}$ between 0.009–0.031 s$^{-1}$. The msfGFP-PBP2 fusion shows less frequent arrests and faster transitions from immobile to persistently moving states, in agreement with its higher fraction of moving molecules (*Figure 1—figure supplement 10A*).

A persistent run of PBP2 terminates either due to an arrest of PBP2 (persistent-to-immobile transition) or due to an unbinding event (persistent-to-diffusive transition). As an upper bound of the unbinding rate, we measured the transition rate from the aggregate bound state (persistent and immobile states) to the diffusive state, $k_{bd}$ (*Figure 2A*). Specifically, we acquired distributions of track lengths for two different imaging intervals of 1 s and 12 s (*Figure 2C*), using the inducible msfGFP-PBP2 fusion with 25 µM IPTG for the higher number of tracks obtained. Track length is limited by bleaching, unbinding, and persistent molecules leaving the field of view. The latter two processes are responsible for the shorter track lengths observed for d$t$ = 12 s. Taking all three processes into account in computational simulations, we obtained an upper limit of the unbinding rate of $k_{bd}$ <0.03 s$^{-1}$, corresponding to a minimum average lifetime of 30 s (*Figure 2—figure supplement 1*).

The rates $k_{pi}$ and $k_{bd}$ then yield the possible ranges of average run lengths of persistently moving molecules between 0.3–1.6 µm, depending on protein fusion and exact unbinding rate. This range is compatible with long tracks of MreB motion observed previously (*van Teeffelen et al., 2011*).

## PBP2 spatial pattern and bound fraction are independent of MreB cytoskeleton

MreB is often regarded as a hub for Rod-complex components (*Errington, 2015*; *Shi et al., 2018*; *Surovtsev and Jacobs-Wagner, 2018*). To determine whether MreB is the substrate of PBP2 binding we treated cells with the putative MreB-polymerization inhibitor A22 (*Gitai, 2005*). A22 treatment (50 µg/ml) strongly reduced both number and size of MreB-msfGFP spots observed in the cell envelope (*Figure 3A–D*, *Figure 3—figure supplements 1–2*) and abolished rotational motion (*Figure 3—videos 1–2*). We observed the same qualitative behavior for a functional GFP-RodZfusion (*Bendezú et al., 2009*) expressed as the sole copy of RodZ (*Figure 3A–D*, *Figure 3—figure supplements 1–2* ). On the contrary, the spotty pattern of mCherry-PBP2 did not change during A22 treatment. Since these images were taken with a long exposure time (1 s), each spot likely represents multiple bound PBP2 molecules. Our results thus indicate that MreB filaments are not the binding substrate for PBP2. We will come back to the role of RodZ further down.

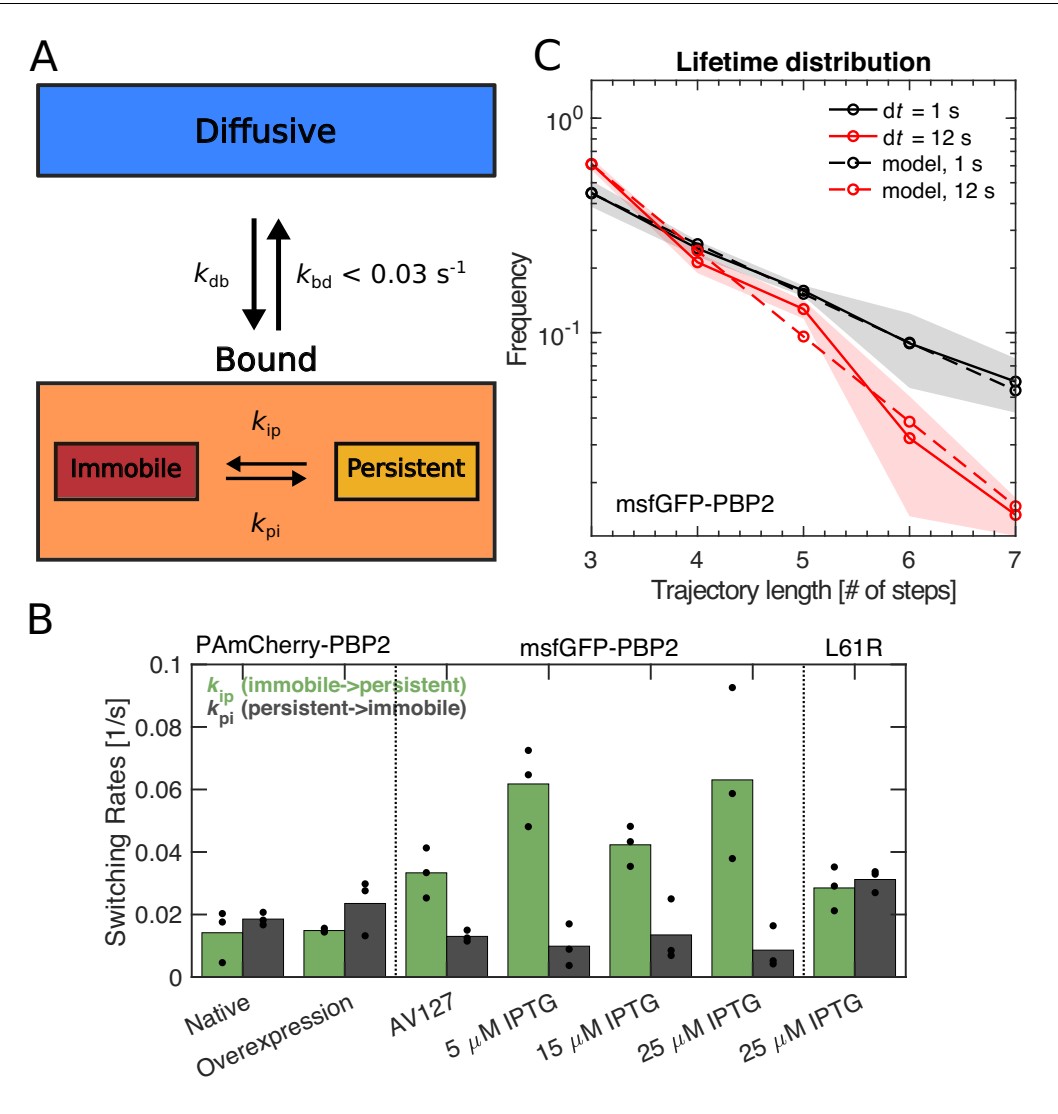

**Figure 2.** PBP2 molecules transition between different dynamic states. **(A)** Diagram illustrating transition rates measured between different motion states. **(B)** Transition rates between immobile and persistently moving states for different protein fusions and expression levels. Circles: biological replicates. **(C)** Fluorescence-lifetime distributions of msfGFP-PBP2 trajectories with imaging intervals of 1 s (black solid line) and 12 s (red solid line). Dashed lines represent a joint fit of the two curves to a model of photobleaching and bleaching-independent track termination, the latter comprising unbinding and persistently molecules leaving the TIR field of view (bleaching probability per frame $p_b = 0.39 \pm 0.08$, apparent track termination rate $k_a = 0.035 \pm 0.007$ s$^{-1}$). Based on a model for persistent motion, we obtained an upper limit of the unbinding rate of $k_{bd} < 0.03$/s (**Figure 2—figure supplement 1**). Shaded region: Standard deviation between at least three technical replicates.

The online version of this article includes the following source data and figure supplement(s) for figure 2:

**Source data 1.** Table containing all data presented in **Figure 2**.

**Figure supplement 1.** Determination of an upper limit of the unbinding rate $k_{bd}$ through simulations.

Next, we measured bound and persistent fractions of PBP2 molecules before and after A22 treatment. The bound fraction remained close to the value of untreated cells for both PAmCherry- and msfGFP fusions (**Figure 3E**; **Figure 3—figure supplement 3**). Yet, the fraction of persistently moving molecules nearly vanished (**Figure 3F**). This is consistent with the arrest of MreB rotation (**Figure 3—video 2**) and with previous bulk measurements of Rod-complex activity (**Uehara and Park, 2008**). To follow the bound fraction during two mass-doubling times (**Figure 3E**), we used an intermediate

concentration of A22 (20 µg/ml), which did not affect growth (*Figure 3—figure supplement 4A*). Together, our findings suggest that MreB polymers are neither the substrate of PBP2 binding nor do they affect the rate of PBP2 binding and unbinding.

MreB depolymerization did also not elicit a rapid change of the diffusion constant of the freely diffusing molecules (*Figure 3E*), suggesting that MreB does not significantly constrain the movement of diffusive PBP2 molecules (*Strahl et al., 2014*). Therefore, if PBP2 binds its substrate from the diffusive state, the locations of PBP2 binding are also independent of MreB.

A22 treatment already demonstrates that PBP2 binding is independent of Rod-complex activity. We confirmed this finding using the PBP2-targeting beta-lactam Mecillinam, which binds covalently to the active site (*Spratt, 1975*). Mecillinam did not cause a reduction of the bound fraction (*Figure 3E*), demonstrating that PBP2 binds its binding target with a moiety different from its active site.

At long times of treatment with Mecillinam or A22 (120 min) the bound fraction increased and the diffusion constant decreased (*Figure 3E*), which coincides and is potentially caused by the loss of normal cell-wall architecture during loss of rod-like cell shape (*Figure 3—figure supplement 4C*), similar to the increase of the bound fraction at sustained low induction levels of msfGFP-PBP2 (*Figure 1—figure supplement 9B*).

## PBP2 binds to its substrate at locations that are independent of MreB localization

To demonstrate that PBP2 binding was indeed independent of MreB filaments or Rod-complex activity as suggested by *Figure 3* we still needed to show that PBP2 molecules interchange between diffusive and bound states during A22 treatment. We already found a low upper bound for the transition rate from bound to diffusive states in non-treated cells ($k_{bd} < 0.03$ s$^{-1}$) (*Figure 2C*), and we expect inverse transitions to occur even more rarely. We therefore used a variant of Fluorescence Recovery After Photobleaching (FRAP) (*Figure 4A*) termed Bound-Molecule FRAP: Instead of measuring fluorescence intensity we measure the bound fraction at different time points, after bleaching almost all molecules at the bottom of the TIR field of view. Right after bleaching, the bound fraction dropped significantly (*Figure 4B*), suggesting that fast diffusing molecules re-entered the observation window within less than half a minute but did not quickly bind their substrate. Within about 2–4 min the bound fraction recovered, yielding a transition rate from diffusive to bound states of $k_{db} = (4.3 \pm 2) \cdot 10^{-3}$ s$^{-1}$. The same experiment in non-treated cells did not reveal recovery of the bound fraction (*Figure 4—figure supplement 1*), likely because bound molecules leave the field of view through persistent motion within less than 1 min (Materials and methods).

Since the bound and diffusive fractions did not change by more than 10% after A22 treatment, transition rates from the diffusive into the bound states must be matched by reverse transitions from the bound to the diffusive state with a rate $k_{bd} = k_{db}(1-b)/b = 0.015 \pm 0.009$ s$^{-1}$, where $b$ is the bound fraction. We confirmed this expectation through independent lifetime measurements of A22-treated cells, similar to those in *Figure 2* (*Figure 4C*), yielding $k_{bd} = 0.018 \pm 0.01$ s$^{-1}$. Since bound fractions are almost identical for untreated and A22-treated cells, we reasoned that binding and unbinding rates $k_{bd}$ and $k_{db}$ are likely also the same in both conditions.

The average lifetime of a bound molecule of about 1 min is 70-fold smaller than the cell doubling time of 70 min (*Figure 1—figure supplement 1C*). Therefore, almost all bound PBP2 molecules observed at any time have undergone multiple transitions from the diffusive to their current bound state. Together with the previous observation that free diffusion is not constrained by MreB filaments we can thus conclude that PBP2 binds its substrate at locations that are determined independently of MreB filaments.

## PBP2 or an unknown low-abundant substrate determines the location of new rod complexes

Two qualitatively different scenarios for the formation of an active Rod complex are conceivable (*Figure 4D*): First, PBP2 could bind to a location in the cell envelope and then recruit MreB filaments directly (through diffusion and capture or through nucleation). Alternatively, MreB filaments could bind the PBP2-binding site independently of PBP2 (however, without actively influencing the sites of PBP2 binding itself, as shown in the previous paragraph).

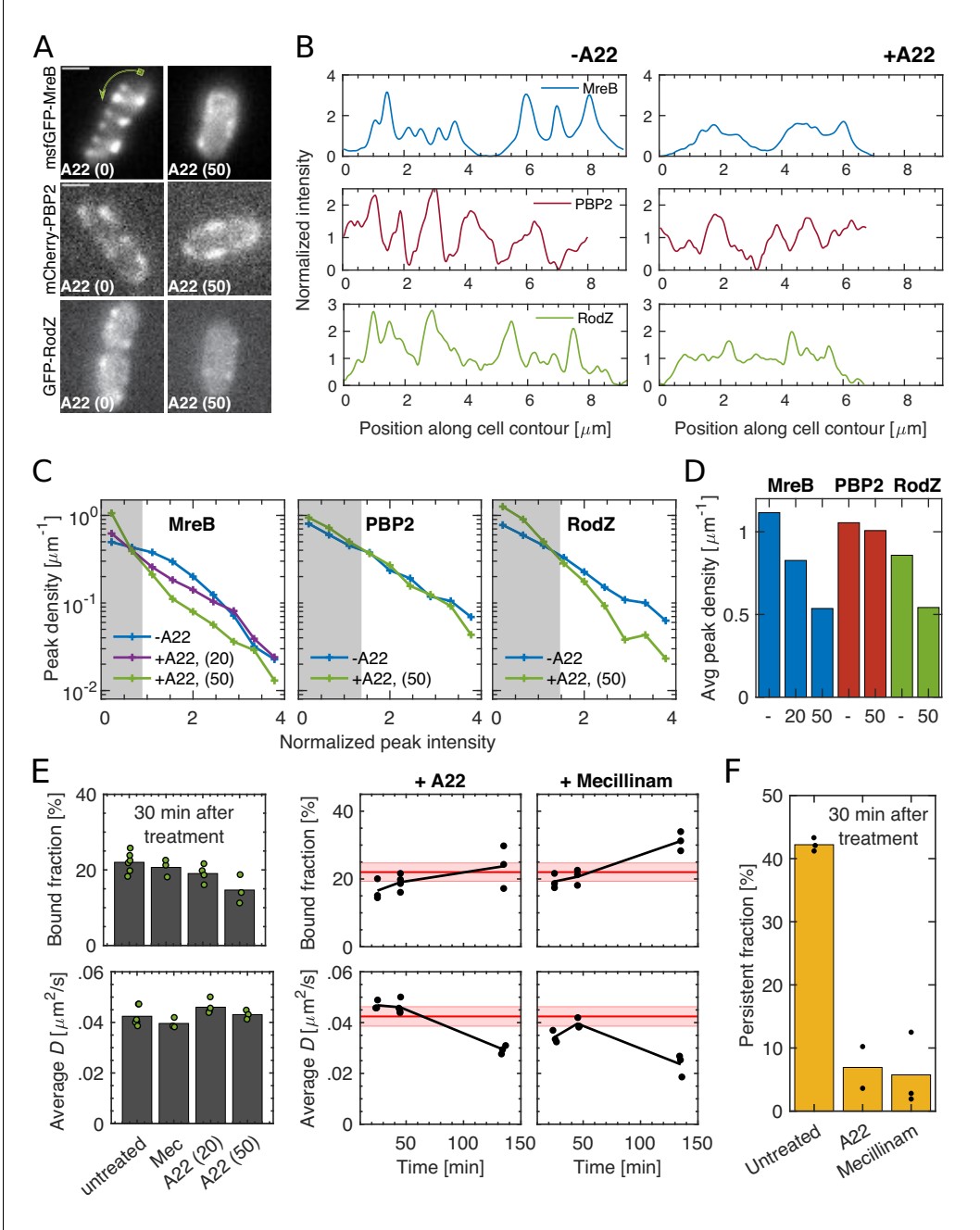

**Figure 3.** Spatial distribution and magnitude of PBP2 bound fraction are independent of MreB cytoskeleton. (A–B) 30 min A22 treatment (50 μg/ml) visibly reduces peak number and intensity of MreB-msfGFP and GFP-RodZ but not of mCherry-PBP2 on cell boundaries, as seen in epi-fluorescence images (A) and in line profiles measured along the cell contour (B), starting from one cell pole at x = 0 as indicated by the green arrow in (A). Image exposure time 1 s. Scale bar 1 μm. (C) Peak density on the cell boundary [1/μm] as function of peak intensity for two A22 concentrations (20, 50 μg/ml). Intensities are normalized by median peak intensity in untreated cells. Gray regions: peaks within noise floor. (D) Density of all peaks above noise floor in (A) for untreated and A22-treated conditions. (E) Left. Bound fraction and diffusion constant of PAmCherry-PBP2 30 min after drug treatment with mecillinam (labeled 'Mec', 100 μg/ml) or A22 (20 or 50 μg/ml). Right. Bound fraction and diffusion constant over time after treatment with A22 (20 μg/ml) or mecillinam (100 μg/ml). Dots indicate technical replicates. Red lines and shaded areas: Average values and standard deviations between biological replicates from untreated cells. (F) Persistent fractions corresponding to the 30 min time point in (E).

The online version of this article includes the following video, source data, and figure supplement(s) for figure 3:

*Figure 3 continued on next page*

*Figure 3 continued*

**Source data 1.** Table containing all data presented in *Figure 3* and *Figure 3—figure supplements 1–4*.

**Figure supplement 1.** Sample fluorescence profiles on cell boundaries as in *Figure 3B*.

**Figure supplement 2.** Spotty patterns of MreB, RodZ and PBP2 in TIRF microscopy.

**Figure supplement 3.** Verification of bound fraction measurements with A22 treated cells carrying the msfGFP-PBP2 fusion.

**Figure supplement 4.** Growth and shape of A22- and mecillinam-treated cells.

**Figure 3—video 1.** MreB-msfGFP imaging.

https://elifesciences.org/articles/50629#fig3video1

**Figure 3—video 2.** MreB-msfGFP imaging of A22 treated cells.

https://elifesciences.org/articles/50629#fig3video2

---

The latter scenario would pose a strong constraint on the binding substrate: It would require that a fraction of binding sites at least as high as the persistent fraction of PBP2 molecules was occupied by MreB filaments. This scenario would thus require that the number of binding sites is not much higher than the number of MreB filaments in the cell envelope, but still greater than the number of bound PBP2 molecules. With a conservative estimate of MreB filaments of about 200 per cell (Materials and methods) and bound molecules of about 10–150 depending on expression and measurement method (*Figure 1—figure supplement 7A*), this constraint would require that the binding substrate is low-abundant. Alternatively, if MreB was recruited through bound PBP2 molecules, binding targets could be highly abundant.

## None of the known Rod-complex components appears as the sole rate-limiting binding substrate for PBP2

To investigate the role of known Rod-complex components different from MreB as potential binding substrates for PBP2, we constructed depletion strains for RodA, RodZ, or MreCD in a background strain expressing either native levels of PAmCherry-PBP2 (for RodZ, MreCD) or overexpressing PAmCherry-PBP2 (for RodA). Without repression, almost all strains showed normal growth rate (*Figure 5—figure supplement 1*), cell shape (*Figure 5A*, *Figure 5—figure supplement 2A-B*), bound fractions (*Figure 5C*), and persistent fractions (*Figure 5D*). Only the RodA-depletion strain showed longer cells and grew slightly slower than the wild type, in agreement with the growth rate of the overexpression strain (*Figure 1—figure supplement 1C*). Furthermore, the RodZ-depletion strain showed slightly higher bound and persistent fractions.

Within 6 hr of depletion, cell shape was perturbed (*Figure 5B*, *Figure 5—figure supplement 2A, B*) , and repressed protein levels were reduced below wildtype levels (*Figure 5—figure supplement 2C*), while PBP2 levels remained close to initial levels (*Figure 5—figure supplement 2D*, *Supplementary file 1b*). Furthermore, growth rates were at most weakly reduced with respect to non-repressed conditions (*Figure 5—figure supplement 1*).

Upon depletion of RodA or MreCD, bound fractions remained constant (*Figure 5C*). According to previous measurements (*Vigouroux et al., 2018*), RodA was likely repressed by at least three-fold below levels of bound PBP2 molecules (Materials and methods), suggesting that RodA is not the main binding substrate. Due to high residual levels of MreC and possibly high levels of undetected MreD it is possible that either or both of these proteins are required for PBP2 binding. However, constancy of the bound fraction despite a 2.3-fold reduction of MreCD levels with respect to MG1655, suggests that MreC and MreD are not rate-limiting for PBP2 binding.

Depletion of RodZ led to a significant reduction of the bound fraction (p-value=0.021). At the same time, the persistent fraction remained high, which is compatible with observations of reduced but continued MreB motion (*Dion et al., 2019*). The drop of the bound fraction already occurred within two hours (*Figure 5—figure supplement 2F*), suggesting that RodZ modulates the rates of PBP2 binding or unbinding.

Despite the strong effect of RodZ depletion, multiple observations suggest that RodZ is not the sole rate-limiting substrate for PBP2 binding: First, the bound fraction increased after 6 hr of depletion, despite a continued low level of RodZ (*Figure 5—figure supplement 2C, F*). Second, the spatial pattern of RodZ but not of PBP2 depends on the presence of MreB filaments (*Figure 3*), suggesting that the localization of PBP2 does also not depend on RodZ. Third, correlations between

spatial patterns of mCherry-PBP2 and GFP-RodZin cells expressing both fusions as sole copies of the respective proteins were low, in contrast to GFP-RodZand MreB-mCherry (*Figure 5—figure supplement 3*; *Alyahya et al., 2009*; *Morgenstein et al., 2015*).

Finally, *ponA* deletion (PBP1a) (*Figure 5*) led to a drop of the bound fraction. Therefore, PBP1a is not required for PBP2 binding but might aid PBP2 binding or stabilize bound PBP2.

In conclusion, RodA and PBP1a are likely not the substrate for PBP2 binding, and we found evidence that neither MreCD nor RodZ are the sole rate-limiting substrate for PBP2.

## MreB-curvature correlations are likely the result of persistent motion

Previously, Ursell et al. observed that MreB filaments were excluded from regions of positive Gaussian cell-envelope curvature such as found at the cell poles, while MreB was enriched in regions of negative Gaussian curvature as found at the inner sides of bent cells (*Ursell et al., 2014*). They concluded that the locations of Rod-complex activity are determined by MreB filaments preferentially localizing to sites of negative Gaussian curvature in rod-shaped cells (*Ursell et al., 2014*). This conclusion is in contradiction to our finding that PBP2 is responsible for the initial localization of new Rod complexes in the cylindrical part of the cell. However, MreB-curvature correlations could also come about indirectly through persistent rotational motion (*Hussain et al., 2018*; *Wong et al., 2017*; *Wong et al., 2019*) or additional mechanisms of polar exclusion (*Kawazura et al., 2017*), without any curvature-based Rod-complex initiation. To resolve this conflict, we reinvestigated MreB-curvature correlations and their potentially different origin.

We followed a very similar approach to *Ursell et al. (2014)*. Specifically, we measured the spatial pattern of MreB-msfGFP (*Ouzounov et al., 2016*) on the two-dimensional cell contour (*Figure 6A-B*) both in filamentous cells, through expression of the division inhibitor SulA (*Bi and Lutkenhaus, 1993*), and in non-filamentous cells growing on agarose pads under the microscope. We obtained the contour curvature of the cell from phase-contrast images using the Morphometrics cell segmentation tool (*Ursell et al., 2017*; *Ursell et al., 2014*). In cylindrical regions of normally growing or filamentous cells with low variations of cell diameter $\sigma$, contour curvature $\kappa$ is a good proxy for Gaussian curvature $G$, with $G = 2\kappa/\sigma$.

First, we measured the enrichment of MreB intensity at the cell contour as a function of local contour curvature, just like (*Ursell et al., 2014*; *Figure 6C* for filamentous cells; *Figure 6—figure supplement 1* for wild-type cells). In quantitative agreement with their data we found enrichment of MreB at negative curvature and depletion at positive curvatures, as present at cell poles.

We then constrained our analysis to the cylindrical part of the cells and found that correlations were reduced by about five-fold at positive curvature values (*Figure 6C*). These findings demonstrate that the pattern of MreB localization is not simply a function of contour or Gaussian curvature. Instead, curvature correlations are qualitatively different at different parts of the cell and dominated by cell poles. On the contrary, correlations between MreB and contour curvature in the cylindrical part of the cell are weak.

In previous work by some of us (*Wong et al., 2017*) we found small but significant correlations between MreB and cell-centerline curvature in mechanically bent cells. We therefore wondered whether residual correlations between MreB and contour curvature in the cylindrical part of the cell observed here were dominated by weak cell bending (*Figure 6A*) likely caused by cells attaching to the glass surface (*Duvernoy et al., 2018*). To that end we restricted our analysis to regions of the cell, where the spatially filtered cell centerline (using a Gauss filter of $\sigma$ = 80 nm) was nearly straight (*Figure 6A*). These regions still showed variations of cell-envelope curvature due to bulges or indentations (*Figure 6F*). However, we did not find any significant correlations between MreB and contour curvature (*Figure 6D*). Therefore, all residual MreB-curvature correlations after removal of poles and septa can be attributed to weak cell bending, while bending-independent bulges and indentations do not affect MreB localization.

We confirmed our findings with two alternative approaches: First, we subtracted from the local contour curvature the curvature contribution due to cell bending (*Figure 6—figure supplement 2A-B*). Residual curvature fluctuations originate from bulges or indentations. Consistently with our observation in straight cell segments, we found no significant correlations between MreB and corrected contour curvature. In an independent approach, we corrected MreB intensity values along the contours for the effect expected from cell-centerline bending (*Figure 6—figure supplement 2C*). Again, we did not find residual correlations after correction. Both of these analyses therefore

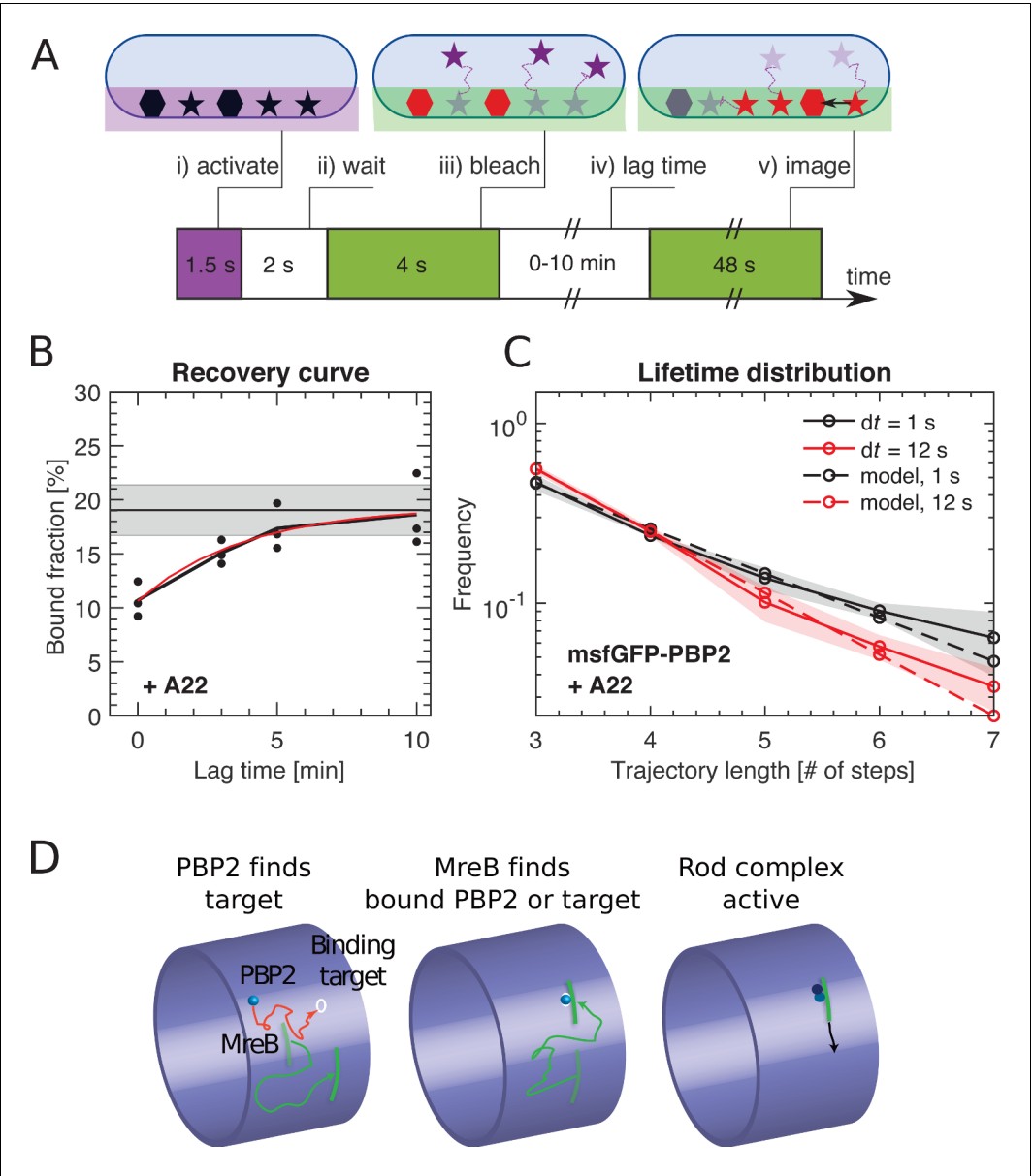

**Figure 4.** PBP2 slowly transitions between diffusive and bound states. **(A)** Bound-Molecule-FRAP reveals rate of PAmCherry-PBP2 binding $k_{db}$: (i) Diffusive (stars) and bound (hexagons) molecules are activated at bottom of cell through TIR illumination. (ii) Most activated diffusive molecules (purple) leave the field of view. (iii) Remaining molecules are bleached (red). (iv) Activated diffusive molecules partially return into the field of view, where they can bind (black arrow). (v) Measurement of bound fraction. **(B)** Bound fraction of PAmCherry-PBP2 in A22-treated (20 μg/ml) cells according to **(A)** at different lag times. Black horizontal line and shaded area: bound fraction without bleaching and standard deviation from technical replicates. An exponential fit in the form $b(t) = a_1 - a_2 \exp[-k_{db} t]$ (red line) yields binding rate $k_{db} = (4.3 \pm 2)*10^{-3}$ s$^{-1}$. Dots represent technical replicates. Shaded area shows standard deviation between six biological replicates. **(C)** Fluorescence-lifetime distributions of msfGFP-PBP2 tracks in A22-treated cells with imaging intervals of 1 s (black solid line) and 12 s (red solid line) yields unbinding rate $k_{bd} = 0.02 \pm 0.01$ s$^{-1}$. Shaded area shows standard deviation between at least three technical replicates. **(D)** Cartoon of suggested Rod-complex initiation: PBP2 (blue) binds to a target site in the cell envelope (white circle) independently of MreB filaments or PBP2 activity. PBP2 or the target site then recruits an MreB filament through diffusion and capture (green) or through nucleation, and also recruits other rod-complex components (magenta). The online version of this article includes the following source data and figure supplement(s) for figure 4:

**Source data 1.** Table containing all data presented in *Figure 4* and *Figure 4—figure supplement 1* .
**Figure supplement 1.** Change in the bound fraction after photobleaching of untreated cells.

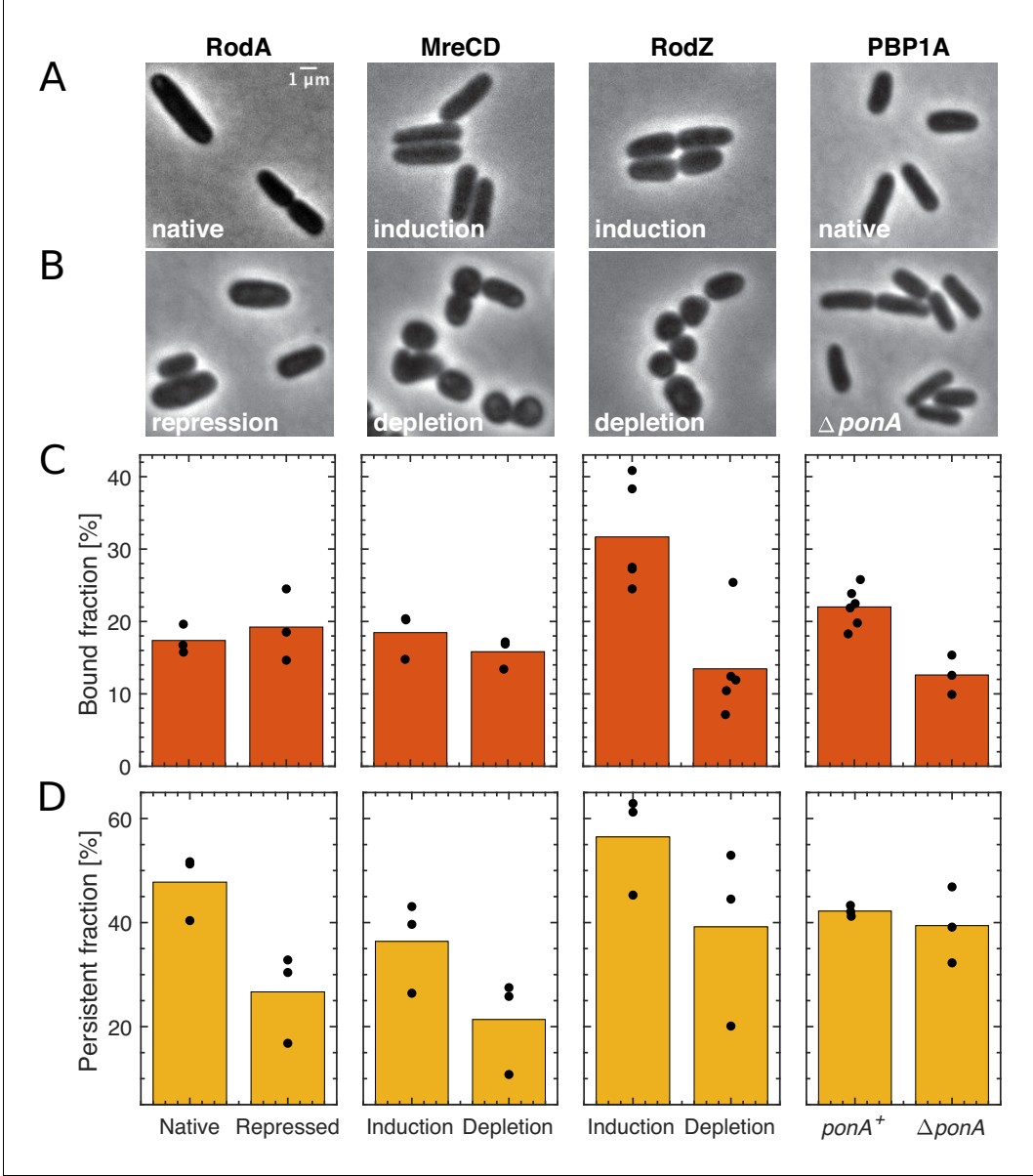

**Figure 5.** The effect of the depletion of Rod-complex components on PBP2 binding and activity. (**A–B**) Cell shape upon near-native expression (**A**) or long-time depletion (**B**) of RodA, MreCD, RodZ, or PBP1a. RodA was repressed for 9 hr through CRISPRi against *mrdAB* operon (coding for PBP2 and RodA) in AV48/pKC128 [P$_{mrdA}$::PAmCherry-PBP2]. Here, PBP2 was 2–5-fold overexpressed from plasmid pKC128 to avoid PBP2 depletion upon *mrdAB* repression. MreCD was depleted for 6 hr in TKL130 Δ*mreCD*/pFB121 [P$_{lac}$::*mreCD*]. RodZ was depleted for 6 hr in TKL130 Δ*rodZ*/pFB290 [P$_{lac}$::*rodZ*]. PBP1a is not essential and was deleted. In all cases except for PBP1A, cells loose rod-like cell shape. (**C–D**) Bound fractions (**C**) and persistent fractions (**D**) of PAmCherry-PBP2 upon expression or depletion of proteins indicated above (**A**). Dots represent biological replicates.

The online version of this article includes the following source data and figure supplement(s) for figure 5:

**Source data 1.** Table containing all data presented in *Figure 5* and *Figure 5—figure supplements 1–3*.
**Figure supplement 1.** Depletion of Rod-complex components shows no effect on growth.
**Figure supplement 2.** Depletion of Rod-complex components.
**Figure supplement 3.** RodZ and PBP2 do not colocalize, while RodZ and MreB do.

strongly support the conclusion that spontaneous cell bending is responsible for all correlations between MreB and contour curvature in the cylindrical parts of normally growing cells.

Previously, we demonstrated that a small bending-induced enrichment of MreB can be explained by persistent rotational motion of MreB filaments (*Wong et al., 2017*), because rotating MreB filaments tend to accumulate at inner regions of bent cells. Our observations are therefore compatible with a model of MreB-independent initiation of Rod complexes through PBP2.

## Diffusing PBP2 molecules do not contribute to Rod-complex activity

It was previously suggested that diffusive PBP2 molecules contribute to cell-wall synthesis (*Lee et al., 2014*). However, if diffusive PBP2 molecules indeed contributed to processive Rod-complex activity, any cross-linking site of a moving Rod complex would have to be found by independent PBP2 molecules through diffusion at a rate equal to the cross-linking rate of up to 15/s. This rate corresponds to the distance between cross links of 2 nm (*Meroueh et al., 2006*) and a speed of PBP2 of 30 nm/s frequently observed (*Figure 1F*).

To test whether PBP2 diffusion was fast enough for the biological cross-linking rate, we conducted computational simulations of freely diffusing PBP2 molecules and measured the average encounter rate between any simulated molecule and a given target site representing a Rod complex (*Figure 7A,B*). We found that the rate between encounters was at least three times lower than the rate of cross-linking, even if a single enzyme had a high chance of returning to the same site and conduct multiple cross-linking reactions on average (*Figure 7C*). Therefore, free diffusion cannot account for physiological cross-linking rates.

As an extension of the model we considered the possibility that PBP2 molecules underwent facilitated diffusion by preferentially diffusing along MreB filaments, which could possibly serve to increase encounter rates between PBP2 and cell-wall-insertion sites. One-dimensional diffusion along filaments indeed increased the encounter rate (*Figure 7D*). However, due to the preferentially circumferential orientation of MreB filaments, facilitated diffusion would also lead to reduced and asymmetric diffusion (*Figure 7E*). On the contrary, we did not observe increased diffusion around the circumference (*Figure 7F*). Therefore, Rod-complex activity requires the stable association between transglycosylase and transpeptidase for multiple cross-linking events. Our findings suggest, that only the persistently moving fraction of PBP2 molecules substantially contributes to cross-linking.

## Discussion

In summary, we found an important role of PBP2 for Rod-complex initiation and persistent cell-wall synthetic activity. New Rod complexes are initiated once PBP2 binds to an immobile substrate in the cell envelope that is different from MreB filaments. Furthermore, we found evidence that none of the known Rod-complex components provides the sole rate-limiting binding substrate for PBP2.

### PBP2 might bind directly to the cell wall

Based on our observations we speculated that PBP2 might bind to the cell wall directly. Support for this viewpoint comes from the diffusive motion of PBP2 molecules. PBP2 diffusion is much slower than diffusion of similar-size membrane proteins (*Kumar et al., 2010*) or of a truncated version of PBP2 (*Lee et al., 2014*), suggesting that PBP2 might weakly bind the cell wall even during diffusion (*Lee et al., 2014*). We found that depletion of RodA, RodZ, and MreCD caused an additional decrease of the diffusion constant (*Figure 5—figure supplement 2E*), similarly to long-term treatment with A22 or Mecillinam (*Figure 3E*). In all cases, Rod-complex activity is inhibited or reduced, which changes cell-wall architecture (*Wang et al., 2012*) and reduces the degree of cross-linking (*Uehara and Park, 2008*). These observations support the model that diffusion is governed by the physical interactions between PBP2 and the cell wall (*Lee et al., 2014*). Interestingly, overexpression of PBP2 also led to a reduction of the diffusion constant (*Figure 1—figure supplement 2*). To determine whether this reduction is due to an alteration of the cell-wall structure or due to a different cause, will require further investigation.

### Cell-wall architecture and not envelope curvature likely provides the signal for Rod-complex initiation

It has been proposed that Rod complexes are recruited to sites of specific cell-envelope curvature based on mechanical properties of MreB (*Ursell et al., 2014*). We found that correlations between

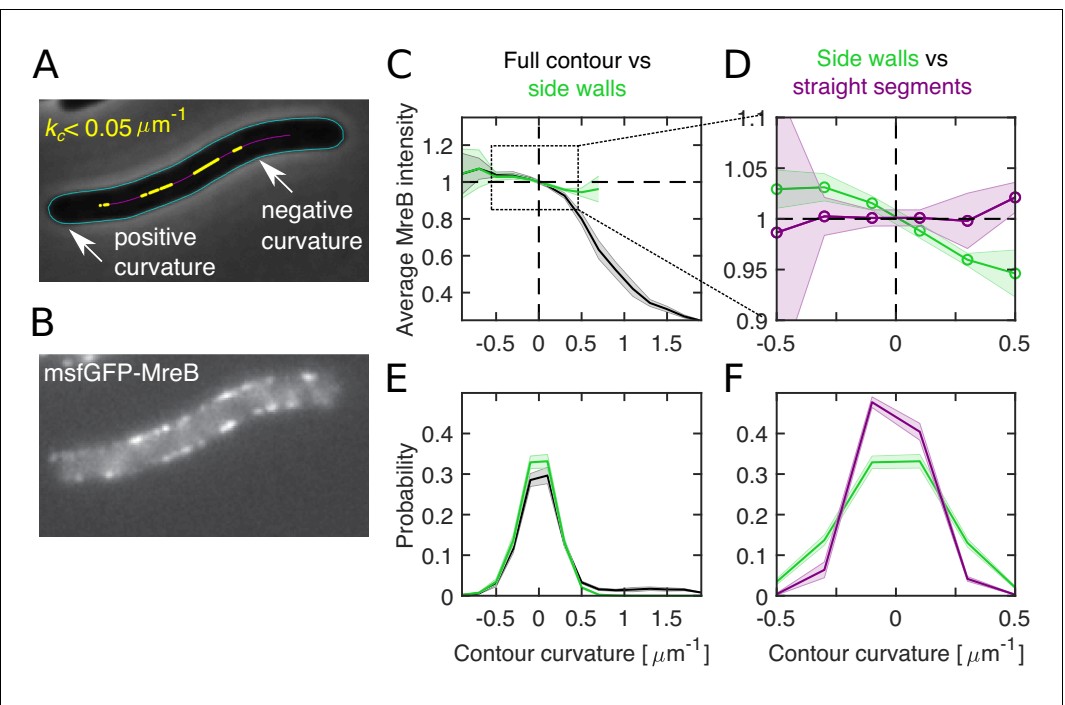

**Figure 6.** Differential MreB-curvature correlations in filamentous cells are due to cell poles and cell bending. (A–B) Phase-contrast image (A) and fluorescence intensity (B) of a representative filamentous *E. coli* expressing MreB-msfGFP and SulA (NO53/pDB192). Contours (cyan) are obtained by computational cell segmentation. Positive contour curvature is found at cell poles, bulges, and outer parts of spontaneously bent regions, while negative curvature is found at indentations and inner parts of bent regions. Straight cell segments (yellow) are defined as regions where the curvature of the spatially averaged centerline (magenta) is smaller than 0.05 μm$^{-1}$. (C–D) Normalized average MreB intensity as a function of local contour curvature. Comparison between correlations obtained from full contours (black) and side walls (green) (C) and from side walls (green) and straight cell segments (magenta) (D). (E–F) Distributions of contour-curvature values corresponding to correlation plots in (C–D). Shaded region: Standard deviation between three biological replicates.

The online version of this article includes the following source data and figure supplement(s) for figure 6:

**Source data 1.** Table containing all data presented in *Figure 6* and *Figure 6—figure supplements 1–2*.
**Figure supplement 1.** Correlations between MreB and contour curvature in WT cells.
**Figure supplement 2.** Loss of correlations between MreB and contour curvature after renormalizing either curvature or intensity for cell bending in filamentous cells.

MreB filaments and cell-envelope curvature in normally growing rods do not require any curvature-dependent initiation of Rod complexes. This observation does not rule out that MreB-curvature correlations in cells of strongly perturbed shape might be influenced by MreB-filament bending or twisting (*Bratton et al., 2018*; *Colavin et al., 2018*). Evidence for motion-independent curvature preferences comes from *C. crescentus* (*Harris et al., 2014*). However, our study as well as previous studies (*Hussain et al., 2018*; *Wong et al., 2017*; *Wong et al., 2019*) suggest that MreB-filament rotation around the circumference are responsible for MreB-curvature correlations in wild-type and filamentous cells. Therefore, the physical signal underlying the spatial pattern of new Rod complexes is likely found in the local architecture of the cell wall, and not, as previously suggested, in the geometry of the cytoplasmic membrane.

## What is the PBP2-binding substrate?

The bound fraction of PBP2 molecules remained nearly constant upon A22 or Mecillinam treatment (*Figure 3*). We thus reasoned that persistently moving and immobile molecules are likely bound to the same substrate. In Gram-negative *E. coli*, active Rod complexes are thought to insert nascent glycan strands in between template strands (*Höltje, 1998*), even if deviations from perfect alignment are reported (*Turner et al., 2018*). During cell-wall insertion, Rod complexes might therefore stay

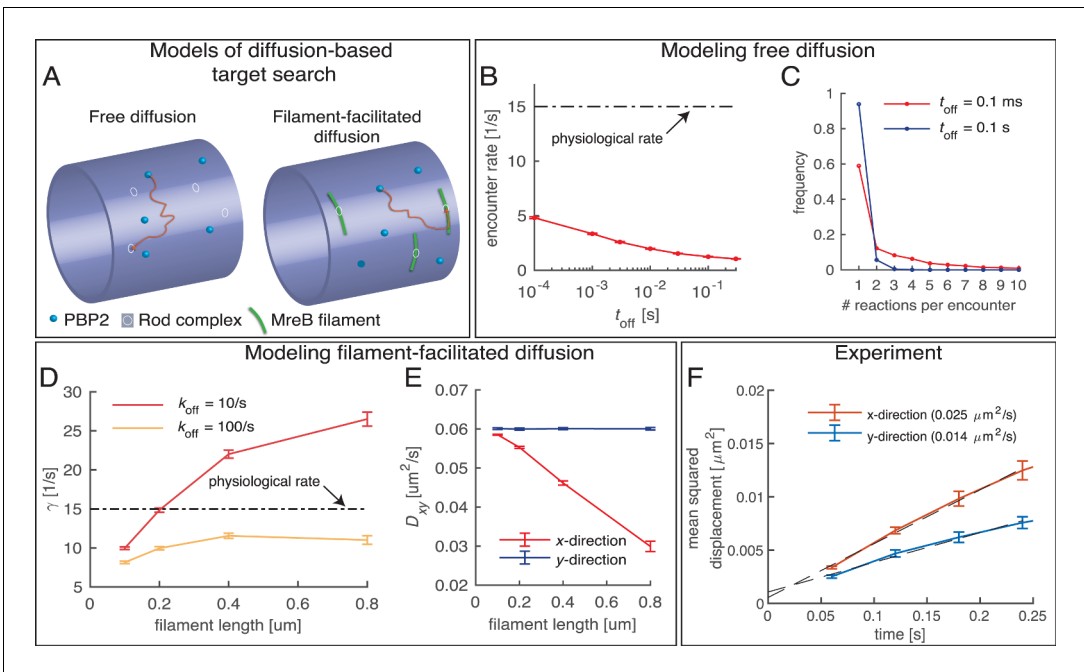

**Figure 7.** Testing a possible role of diffusive PBP2 for cross-linking. (A) Cartoon of PBP2 finding the target site of a 'rod complex' through free diffusion (left) or filament-facilitated diffusion (right). (B) The average encounter rate between any of 100 freely diffusing PBP2 molecules and a given rod-complex site as a function of the unknown latency time $t_{off}$ (the duration for which a single PBP2 enzyme is inactive after a cross-linking reaction) (red) in comparison to the physiological cross-linking rate (dashed-dotted line) (C) Distribution of the number of successive cross-linking reactions conducted by the same PBP2 molecule at the same rod-complex site for two different latency times. (D–E) Facilitated diffusion along circumferentially oriented filaments centered at every rod-complex site increases the encounter rate (D) and renders diffusion asymmetric (E). (F) Diffusion of PAmCherry-PBP2 along the long axis of the cell (x-direction) is faster than around the circumference (y-direction), suggesting that PBP2 does not undergo facilitated diffusion along circumferentially oriented filaments. Reduced diffusion around the circumference is possibly caused by out-of-plane motion.

connected to the local cell wall through associations between PBP2 and a template strand, independently of enzymatic activity. In the future, it will be interesting to study potential interactions at the molecular level. These might then also reveal structural features of the cell wall potentially responsible for stable PBP2 association.

## What determines the rates of binding and unbinding?

PBP2 molecules transition only slowly between bound and diffusive states (*Figure 2*). Possibly, PBP2 is found in two different molecular states that facilitate stable binding or allow for diffusive motion – either through conformational change or through interaction with an unknown interaction partner. Depletion experiments with RodA through CRISPRi suggest that RodA is not involved in this process, since RodA levels were likely repressed below the levels of bound PBP2. Depletion experiments with MreCD and RodZ suggest that none of these proteins constitutes the sole rate-limiting binding substrate. However, residual protein levels upon depletion were too high to rule out an important and possibly essential role for PBP2 binding. Specifically, RodZ depletion led to a reduction of the bound fraction, suggesting that RodZ directly modulates PBP2 binding or the stability of the bound form of PBP2. This is compatible with previous observations of RodZ-PBP2 interactions (*Bendezú et al., 2009*; *Morgenstein et al., 2015*). It will thus be interesting to investigate the role of RodZ for PBP2 binding or unbinding in more detail in the future.

## How do MreB filaments 'find' the binding substrate?

We reasoned that MreB filaments must either be recruited to bound PBP2 molecules or to an unknown, low-abundant binding substrate. To find its target, MreB filaments could explore the cell

envelope through rapid diffusive motion previously overlooked. Alternatively, the binding target could nucleate new MreB filaments. Other Rod-complex components such as RodZ could facilitate this process. This latter hypothesis is supported by the recent observation that the hyperactive PBP2 (L61R) mutant described above causes more and shorter MreB filaments (*Rohs et al., 2018*).

### High residual rod-complex activity upon PBP2 overexpression suggests that PBP2 activates the Rod complex

We observed that the fraction of bound PAmCherry-PBP2 molecules decreased by about two-fold upon three-fold increase of enzyme levels. Together with a nearly constant fraction of persistently moving molecules, this suggests that the number of active enzymes increases about two-fold (*Supplementary file 1a*). Therefore, PBP2 appears to be an important limiting factor for Rod-complex activity. This finding is consistent with the previous observation that PBP2 activates RodA (*Rohs et al., 2018*).

For the msfGFP-fusion we observed that the decrease of the bound fraction was roughly inversely proportional to expression level up to 5 µM IPTG (*Supplementary file 1a*). However, while this level of induction lead to wild-type levels of enzymes on average, it still led to a time-dependent change of cell shape and bound fraction. The quantitative interpretation of this result is therefore difficult.

Both fusions suggest that PBP2 abundance might not be the sole limiting factor for PBP2 binding and Rod-complex activity, and we already found two qualitatively different limiting factors: RodZ has a strong effect on PBP2 binding, possibly by facilitating binding or by stabilizing the bound fraction, while RodA and MreCD have seemingly no effect on binding but an important limiting role for persistent motion.

### Rod-complex activity and cell shape

Despite the substantial residual degree of persistent motion upon depletion of MreCD, RodA, or RodZ, the presence of all these components is required to stably maintain rod shape. A recent paper suggested that cell diameter in *E. coli* is partially determined by the spatial density of active Rod complexes (*Dion et al., 2019*). The model is compatible with the fold-change of persistently moving PBP2 molecules and cell-diameter changes observed during MreCD and RodZ depletion. However, during RodA-depletion this simple model does not apply: Due to overexpression of PBP2, the total number of persistently moving PBP2 molecules per cell is likely as high as in the wildtype – even during RodA depletion. In the future, it will therefore be interesting to study how the stoichiometry of different Rod-complex components affects cell-wall insertion and cell shape.

## Materials and methods

**Key resources table**

| Reagent type (species) or resource | Designation | Source or reference | Identifiers | Additional information |
|---|---|---|---|---|
| Strain, strain background (*E. coli*) | TKL130 | (*Lee et al., 2014*) | MG1655 *mrdA::PAmCherry-mrdA* | |
| Strain, strain background (*E. coli*) | EW07 | This work | TKL130 Δ*mreCD*, pFB121 | Materials and methods, Strain construction |
| Strain, strain background (*E. coli*) | EW49 | This work | TKL130 Δ*rodZ*, pFB290 | Materials and methods, Strain construction |
| Strain, strain background (*E. coli*) | AV48 | This work | *186attB::Ptet-dcas9, mrdA::rPAmCherry-mrdA* | Materials and methods, Strain construction |
| Strain, strain background (*E. coli*) | NO53 | (*Ouzounov et al., 2016*) | MG1655 *mreB-msfGFP$^{sw}$* | |
| Strain, strain background (*E. coli*) | TU230(attLHC943) | (*Rohs et al., 2018*) | MG1655 *mrdA::aph* (P$_{lac}$::*msfgfp-mrdA*) | |
| Strain, strain background (*E. coli*) | TU230(attLPR122) | (*Rohs et al., 2018*) | MG1655 *mrdA::aph* (P$_{lac}$::*msfgfp-pbpA*(L61R)) | |

*Continued on next page*

*Continued*

| Reagent type (species) or resource | Designation | Source or reference | Identifiers | Additional information |
|---|---|---|---|---|
| Strain, strain background (*E. coli*) | S352 | This work | MG1655 Δ*ponA*::*aph* | Materials and methods, Strain construction |
| Strain, strain background (*E. coli*) | FB83 | (*Bendezú and de Boer, 2008*) | MG1655, *lacIZYA*::frt, *mreB-mCherry*$^{SW}$ *yhdE*::frt | |
| Strain, strain background (*E. coli*) | AV07 | (*Vigouroux et al., 2018*) | MG1655 *mrdA*::*mcherry-mrdA* | |
| Strain, strain background (*E. coli*) | AV127 | This work | 186::dCas9, *mrdA*::sfgfp-mrdA | Materials and methods, Strain construction |
| Strain, strain background (*E. coli*) | FB60 (iFB273) | (*Bendezú and de Boer, 2008*) | MG1655 *lacIZYA*::frt, *rodZ*::*aph*, P*lac*::*gfp-rodZ* | |
| Strain, strain background (*E. coli*) | S504 | This work | FB83, *rodZ*::*aph*, $P_{lac}$::*gfp-rodZ* | Materials and methods, Strain construction |
| Strain, strain background (*E. coli*) | S505 | This work | AV07, *rodZ*::*aph*, $P_{lac}$::*gfp-rodZ* | Materials and methods, Strain construction |
| Chemical compound, drug | Mecillinam | Sigma-Aldrich, #33447 | | |
| Chemical compound, drug | A22 | Cayman Chemical #15870 | | |
| Software, algorithm | Trackmate | (*Tinevez et al., 2017*) | | |
| Software, algorithm | Morphometrics | (*Ursell et al., 2017*) | | |
| Software, algorithm | Spot-On | (*Hansen et al., 2018*) | | |

## Strain construction

All strains, plasmids, and primers can be found in *Supplementary file 3*.

EW07: To obtain the MreCD depletion strain EW07 (TKL130 Δ*mreCD*, pFB121) we introduced pFB121 (*Bendezú and de Boer, 2008*) into TKL130. We replaced the *mreCD* gene by a kanamycin resistance gene (amplified from pKD13 with primer DmreC_fw and DmreC_rv) using the λ-Red mediated recombineering system expressed from pTKred (*Kuhlman and Cox, 2010*). The resulting strain was then grown at 42°C to cure the plasmid pTKred. We verified the deletion of *mreCD* by PCR.

EW49: For the RodZ depletion strain EW49 (TKL130 Δ*rodZ*, pFB290) we first introduced pFB290 (*Bendezú et al., 2009*) into TKL130. In the resulting strain we deleted *rodZ* by a P1 transduction with lysate made from the Δ*yfgA* strain from the Keio collection yielding strain EW49.

AV48: We constructed a strain AV48 (186::P$_{tet}$-*dcas9*, *mrdA*::r*PAmcherry-mrdA*) that allows repressing the native mrdAB operon using CRISPRi and a guide RNA targeting the codon-modified ORF of r*PAmcherry* (*Vigouroux et al., 2018*), where 'r' denotes the codon modification (repressible). To construct strain AV48 we used allelic exchange on the strain LC69 (186attB::P$_{tet}$-*dcas9*) (*Cui et al., 2018*). To that end, we first electroporated the plasmid pAV25 (*Vigouroux et al., 2018*) into LC69 and selected for growth on chloramphenicol. Then we removed the plasmid backbone by transforming the cells with pAV10 (*Vigouroux et al., 2018*). The resulting strain was then grown at 42°C to cure the plasmid pAV10.

For RodA repression, we transformed AV48 with the CRISPR plasmid pcrRNA G20-R20 (*Vigouroux et al., 2018*) and pKC128 (P$_{mrdA}$-*PAmcherry-mrdA*, non-repressible version) to counter repression of native PBP2.

S352: For the *ponA* deletion strain we deleted *ponA* in TKL130 by a P1 transduction with lysate made from the Δ*ponA* strain from the Keio collection yielding strain S352.

AV127: The strain AV127 was obtained with the allelic exchange procedure described in *Vigouroux et al. (2018)*. LC69 (*Cui et al., 2018*) was electroporated with the suicide plasmid pAV101, then clones that integrated the plasmid by recombination were screened for by PCR. We then transformed these clones with pAV10 (*Vigouroux et al., 2018*) in a medium containing diacetyl-phloroglucinol (DAPG) to induce backbone excision. Finally, clones containing the desired

scarless insertion were identified by PCR. To make pAV101, we amplified the msfGFP-PBP2 fusion from pHC943 (*Cho et al., 2016*) with primers V396 and V397, the backbone from pSW23t with V171 and V394, and an homology region from MG1655's chromosome with V180 and V395. These three fragments were then assembled together by Gibson assembly. We checked the sequence of the fusion gene by Sanger sequencing.

S504: To construct a strain expressing both MreB-mCherry and GFP-RodZ fusions, we used FB83 (*mreB-mCherry*) and performed P1 transduction with lysate made from FB60(iFB273) (Δ*rodZ*, *Plac:: gfp-rodZ*) (*Bendezú et al., 2009*) first selecting on ampicillin for Plac::*gfp-rodZ* then a second transduction with the same lysate and selecting on kanamycin for deleting the native *rodZ*.

S505: To construct a strain carrying both mCherry-PBP2 and GFP-RodZ fusions, we used strain AV07 (*mCherry-mrdA*) and performed P1 transduction with lysate made from FB60 (iFB273) (Δ*rodZ*, *Plac::gfp-rodZ*) (*Bendezú et al., 2009*) first selecting on ampicillin for Plac::*gfp-rodZ* then a second transduction with the same phage lysate and selecting on kanamycin for deleting native *rodZ*.

## Bocillin labelling

PBP2 levels of strains MG1655, TKL130 and TKL130/pKC128 were measured by a Bocillin-binding assay as described similarly in *Cho et al. (2016)* and *Kocaoglu et al. (2012)*. We performed the quantification in parallel on two identical cultures grown from independent colonies of each strain. We grew strains overnight in LB at 37° C. We then washed the cells in LB, diluted them 1/200 in LB and grew them at 37° C until an OD600 of ~0.4. We washed 1.8 ml of the culture in PBS, resuspended in 200 µl PBS and kept cultures on ice. We disrupted cells by sonication (FB120, Fisher Scientific) and centrifuged them for 15 min at 4° C (21,000 g). We subsequently resuspended the pellet corresponding to the membrane fraction in 50 µl PBS containing 15 µM fluorescently labelled Bocillin-FL (Invitrogen). Membranes were incubated at 37° C for 30 min and washed once in 1 ml PBS. We centrifuged the membranes for 15 min (21,000 g) and resuspended them in 50 µl PBS. We measured the protein concentration of each sample with a colorimetric assay based on the Bradford method (#5000006, Bio-Rad) and loaded equal amounts of protein mixed with 4X Laemmli buffer onto a 10% polyacrylamide gel. We visualized the labelled proteins with a Typhoon 9000 fluorescence imager (GE Healthcare): excitation at 488 nm and emission at 530 nm. We quantified the relative amounts of PBP2 in each sample by quantifying the grey values of each lane in ImageJ (*Schneider et al., 2012*).

## Sample preparation and imaging conditions

Cells were grown overnight at 37°C in LB medium and then washed and diluted at least 1:500 in M63 minimal medium (*Miller, 1972*) supplemented with 0.1% casamino acids, thiamine ($5 \cdot 10^{-5}$%), glucose (0.2%) and MgSO4 (1 mM) (here referred to as minimal medium) and grown to early exponential phase (maximum OD600 of 0.1) at 30°C. If strains carried antibiotic resistance we added the respective antibiotics to the medium (carbenicillin (100 µg/ml), kanamycin (50 µg/ml), chloramphenicol (30 µg/ml), all Sigma-Aldrich). Inducers were added as indicated below. For microscopy, cells were spotted on 1% agarose pads (UltraPure Agarose, Invitrogen) of minimal medium, without any antibiotics. Drugs targeting MreB or cell-wall synthesis and inducers were added as indicated below.

For technical replicates, we grew cell cultures to exponential phase in independent flasks starting from the same saturated overnight culture. For biological replicates, we grew overnight cultures from different bacterial colonies.

Cells were imaged in a custom-built temperature-controlled chamber at 29°C. Prior to imaging the cells were incubated 10–20 min on the microscope to equilibrate sample temperature and minimize drift. In experiments with drugs [mecillinam (Sigma-Aldrich, #33447), A22 (Cayman Chemical #15870)] cells were spotted on agarose pads containing the antibiotic at the concentration of 100 µg/ml (mecillinam), or 20–50 µg/ml (A22). For treatments longer than 20 min cells were incubated in liquid culture with the respective antibiotic at the same concentration prior to imaging. The indicated incubation times denote the total incubation time in liquid and on the pad.

In experiments where we depleted MreCD and RodZ (strains EW07 and EW49 respectively) we grew cells in the presence of 50 or 100 µM IPTG respectively (EUROMEDEX #EU0008-B) and carbenicillin both during overnight growth and during regrowth as described above. We then washed the cells three times, diluted them and grew them in minimal medium in the absence of IPTG for 6 hr.

In experiments where we repressed the *mrdAB* operon (strain AV48/pcrRNA G20-R20/pKC128) we grew cells in the presence of kanamycin and chloramphenicol both during overnight growth and during regrowth as described above. We added anhydrotetracycline (aTc) (100 ng/ml, Acros Organic) upon dilution in minimal medium to induce the repression. We grew cells for 9 hr prior to imaging both in the presence and absence of aTc.

In experiments where we imaged msfGFP-PBP2, TU230(attLHC943) cells are plated on and grown overnight in minimal media containing 25 µM IPTG. For the regrowth from saturated culture, we used minimal media containing the indicated IPTG concentration. We grew cells for 6, 8 or 10 hr to early exponential phase at 30°C. For the measurement of auto fluorescence, we grew MG1655 in the same way but without the addition of an antibiotic or IPTG.

In experiments where we imaged GFP-RodZ, we used 100 µM IPTG.

For MreB-curvature correlation measurements we grew NO53 or NO53/pDB192 overnight at 37°C in LB medium. We diluted the cultures by 1:500 and grew them at 30°C in LB medium for 2 hr. We then washed the cultures and diluted them 1:200 in MOPS rich medium (MOPS EZ Rich Defined Medium Kit. Cat.No. M2105, TEKnova) and grew them at 30°C for an additional 4 hr, such that the culture density remained below an OD600 of 0.1. The growth media of NO53/pDB192 cells contains carbenicillin at all times until the cells are harvested for microscopy. Cell division is inhibited by adding 1 mM IPTG in liquid culture inducing *sulA* expression 30 min prior to imaging. Cells were placed on 1% agarose pads of the same media without any antibiotics but containing the same amount of IPTG for filamentation. Images were taken right after placing cells on a pad (NO53) or after maintaining cells for 30 min on the pad at the same 30°C (NO53/pDB192).

## Microscopy

Single particle tracking of PAmCherry-PBP2 and msfGFP-PBP2 was performed on a custom-designed fluorescence microscope (here referred to as microscope 1) in TIRF (total internal reflection fluorescence) mode. The microscope was equipped with a 100x TIRF objective (Apo TIRF, 100x, NA 1.49, Nikon), three laser lines: 405 nm (Obis, Coherent), 488 nm (Sapphire, Coherent), 561 nm (Sapphire, Coherent), a dichroic beamsplitter (Di03-R488/561-t3−25 × 36, Semrock), an edge filter for PAmCherry imaging (BLP02-561R-25, Semrock), and a laser-line filter (NF561-18, Thorlabs). Shuttering of the 488 nm laser was controlled with an acusto optic tunable filter (AA Optoelectronics) or with shutters (Uniblitz, LS3 and TS6B, Vincent Associates). The 405 nm laser was controlled directly via an USB interface. Images were acquired with an EMCCD camera (iXon Ultra, Andor). All components were controlled and synchronized using µManager (*Edelstein et al., 2010*).

For high-frequency imaging with PAmCherry-PBP2, images were acquired with exposure time and intervals of 60 ms for a duration of 1 min. A weak UV-laser pulse (405 nm, 100–200 ms) was provided before the acquisition and every 200 frames during the measurements to activate new fluorophores. To obtain low variability of the bound fraction of molecules between replicates (standard deviation <5%) we aimed to generate about 1000 tracks per technical replicate by taking 3–10 movies per replicate.

Low-frequency imaging was conducted with an exposure time of 1000 ms and with imaging intervals of 3.6 s or 1 s for a duration of 3.5 min or 3 min, respectively. We activated fluorophores once before the acquisition for 100–200 ms. In this way we collected 7–9 movies per replicate yielding about 100 tracks of at least four time steps, which resulted in variability of the persistent fraction of standard deviation <10%. This way, the imaging time did not exceed 30 min per replicate.

Single molecule tracking with cells carrying the msfGFP-PBP2 fusion requires a photo-bleaching phase prior to image acquisition. For high-frequency imaging, the sample was exposed to 488 nm laser in epi mode in order not to bias our analysis towards diffusive molecules as exposure to light in TIR mode would predominantly lead to a loss of bound molecules. After photo-bleaching, we immediately switched to TIR mode for image acquisition. Images are taken with 60 ms exposure time and intervals for a duration of 1 min as for the PAmCherry fusion. As for PAmCherry-PBP2, we aimed to generate about 1000 tracks per replicate.

For low-frequency imaging of msfGFP-PBP2, the sample is first photo-bleached in TIR mode with high laser power. This step is followed by image acquisition while using a reduced laser power with an exposure time of 1000 ms and imaging intervals of 3.6 s for a duration of 4.5 min. Bleaching times for both cases were adjusted according to the PBP2 levels and it varies between 5 to 30 or 2 to 10 s

for high and low frequency imaging respectively. As for PAmCherry-PBP2, we aimed to generate >100 tracks per replicate.

For measurements of cell shape, average fluorescence intensity, MreB-curvature correlations, and MreB rotation we used two inverted epi-fluorescence microscopes, an Eclipse Ti (Nikon) microscope (microscope 2) or a DeltaVision Elite (GE Healthcare) microscope (microscope 3). Microscope two is equipped with a 100x phase contrast objective (CFI PlanApo LambdaDM100 × 1.4 NA, Nikon), a solid-state light source (Spectra X, Lumencor Inc, Beaverton, OR), a multiband dichroic (69002bs, Chroma Technology Corp, Bellows Falls, VT), and excitation (485/25) and emission (535/50) filters. Images were acquired using a sCMOS camera (Orca Flash 4.0, Hamamatsu, Japan) with an effective pixel size of 65 nm. Microscope three is equipped with a 100x phase contrast objective (UPlanSApo 100X NA 1.4, Olympus), a multi-band dichroic beamsplitter (DAPI-FITC-mCh-Cy5) with excitation (475/28) and emission (525/48) filters, and a sCMOS camera (DV Elite, PCO-Edge 5.5) with an effective pixel size of 65 nm.

For measurements of fluorescence on cell boundaries, contour curvature, and MreB-msfGFP tracks, we focused on cells based on the phase-contrast signal. To track MreB-msfGFP spots moving at the bottom of the cell, we moved the focal plane 250 nm below the central plane of cells. Images were taken every 1 s for a duration of 120 s. We generated images of about 100 cells per replicate.

## Cell segmentation

Cell boundaries were detected from phase contrast microscopy images on microscope 2 or three using the MATLAB based cell segmentation tool Morphometrics (SimTK) (*Ursell et al., 2017*). The cell poles and the cell centerline were identified using the MicrobeTracker package (*Sliusarenko et al., 2011*). The spacing of subsequent points along the cell centerline was chosen as 0.5 pixels (32 nm). Cell width was measured as the median of all local widths.

A smoothened centerline (x- and y-coordinates Gauss-filtered, $\sigma$ = 3.5 steps) was used for a cell-internal orthogonal coordinate system, specifically to determine the local orientation of the cell.

## PBP2 tracking

All images taken on microscope one were analyzed using custom Matlab code (available as source code file). First, we segmented bright field images using a semi-automated approach based on standard image processing tools to separate regions containing cells from background regions. For spatially separated cells this also allowed us to obtain a coordinate system for each cell.

PAmCherry-PBP2 spots in fluorescence images were identified as the local maxima of the denoised (bandpass filtered) images with intensity 3.5 or 2 (for fast and slow imaging, respectively) times higher than background (bpass and pkfnd functions from https://site.physics.georgetown.edu/matlab/code.html). Sub-pixel resolution was achieved by finding the center of a two-dimensional Gaussian fitted to the intensity profile of each spot. We discarded peaks with a poor (residuals of fit) or broad (standard deviation of fit) Gaussian profile. Spots in subsequent frames were then connected into raw trajectories if their distance was below a threshold distance that is consistent with diffusion (high-frequency imaging) or persistent motion (low-frequency imaging) (*Crocker and Grier, 1996*). Threshold distances were 600 nm for imaging intervals of $\tau$ = 60 ms, 112.5 nm for $\tau$ = 1 s, and 225 nm for $\tau$ = 3.6 s. If tracking of a particle lead to a situation where a particle can be connected to more than one possible peak in the next frame we discontinued the tracking for this trajectory. We checked that low-frequency trajectories lie in cells by comparing fluorescent images and brightfield images. msfGFP-PBP2 images from fast frequency imaging are analyzed in the same way except that peaks which have intensity 1.6 times higher than background are selected.

msfGFP-PBP2 spots from low frequency imaging were identified and tracked in the following way: Peaks were preselected as local maxima in the bandpass-filtered image (using a Laplacian of Gaussian filter with $\sigma$ = 1.8 pixels and pkfind, as above). We only considered local maxima in regions of the 2% highest intensity values that are 4-connected and contain at least three pixels each. We only consider peaks with nearest-neighbor distance higher than 3.5 pixels.

For further peak selection and sub-pixel localization we considered denoised raw images (using a Gauss filter with $\sigma$ = 0.4 pixels). We selected peaks with a ratio of peak signal to local background noise higher than 3. Local background intensity and noise are respectively the average and standard deviation of the values of the pixels at a distance included between 3 and 4 pixels of the center of

the peak. To avoid very low intensity peaks we also discarded peaks with absolute signal (<2000 counts for 1 s exposure, <1200 counts for 60 ms exposure), and we remove the 1% of the peaks with highest intensities, as those likely represent clusters of molecules. Sub-pixel resolution was achieved by fitting a two-dimensional Gaussian to every selected peak. Peaks with standard deviation >3 or with a ratio of the average absolute value of the residuals divided by peak intensity >0.2 are removed.

Tracking was performed using the same code from *Crocker and Grier (1996)* with a maximum displacement between subsequent frames of 4 pixels (both for slow and fast tracking). For lifetime measurements with a time interval of 12 s, we increased to maximal displacement to six pixels.

For the comparison between different datasets we used Student's t-test for bound fractions, persistent fractions, and diffusion constants where appropriate. P-values are indicated in supplemental datasets corresponding to the different figures.

## MreB tracking

Fluorescence images were analyzed using a custom Matlab code: Images were first filtered in both space and time using a three-dimensional Savitzky-Golay filter with a filter size of 3 pixels in *xy*-directions time three points along the temporal dimension. Images were subsequently de-noised once more using a 2D-Gauss filter ($\sigma$ = 0.5 pixels).

Images were subsequently rescaled by a factor of 5 using spline interpolation to achieve sub-pixel resolution. MreB spots were detected as local maxima inside the cell boundary obtained by segmentation.

The local maxima were connected to construct raw trajectories based on their distance at consecutive time points (*van Teeffelen et al., 2011*) with a maximal displacement during subsequent time frames of 2 pixels.

## Curvature analysis and MreB-curvature correlations

After obtaining the cell contours and the two poles of each cell we first computed the contour curvature all along the cell boundary by fitting a polygon to every contour point and its two neighboring points (MATLAB function LineCurvature2D, http://www.mathworks.com/matlabcentral/). Negative curvature values correspond to indentations and positive curvature values to bulges and poles. To eliminate the influence of noise the contour curvature $c$ is then obtained by Gauss-filtering the raw curvature values ($\sigma$ = 2 steps, corresponding to 65 nm).

To obtain MreB-msfGFP intensities along the cell boundary we first smoothened raw MreB-msfGFP images using a 2D-Gauss filter ($\sigma$ = 0.5 pixels). For interpolation of the GFP images at points close to the cell boundary we then corrected the contour coordinates $r_i$ for systematic shifts between the phase-contrast-based cell contours and the GFP-intensity peaks corresponding to MreB filaments. Specifically, we obtained the corrected contour coordinates $r_i^c$ from the Morphometrics-based contour coordinates $r_i$ according to

$$r_i^c = r_i + \Delta r - r_{\perp i} s$$

Here, the first term $\Delta r = (-0.35, -0.78)$ pixels is a microscope-dependent displacement vector that accounts for a systematic shift between phase-contrast and GFP images. The second term shifts the contour by an amount $s$ = 140 nm inward (along a vector normal to the cell boundary $r_{\perp i}$), such that the corrected contour passes through the positions of the GFP intensity peaks that correspond to the positions of MreB filaments on the sides of the cell. By convention, $r_{\perp i}$ always points outward from the cell center. To make sure to take MreB-msfGFP peaks into account even if they are slightly displaced from the corrected boundary, GFP intensity values $I_i$ are then obtained by linear interpolation of the smoothened MreB-msfGFP image at the positions of the corrected contour and at two other points perpendicular to the boundary and spaced 0.5 pixel inward an outward:

$$I_i = \frac{1}{3}\left[ I(r_i^c - r_{\perp i}\delta) + I(r_i^c) + I(r_i^c + r_{\perp i}\delta) \right],$$

where $\delta$ = 65 nm. Subsequently, intensity values are normalized by the average taken over the full contour, the side walls, or the straight cell segments, respectively, depending on the analysis. To obtain the MreB enrichment as a function of curvature, the curvature values are binned and MreB

intensities corresponding to those curvature vales are averaged. Subsequently, intensity values are normalized by the average intensity found close to zero curvature:

$$I_i^c = I_i / \langle I_i | -0.05/\text{um} < c_i < 0.05/\text{um} \rangle.$$

If a bin contains less than at least 0.1% of the data points per replicate, it is not displayed in the figure.

To remove the cell poles and potential septa from the analysis, we removed 2 um from each cell pole (according to the distance along the centerline) and 650 nm from the middle of the cell. The large distance from the poles was chosen, since the automated software does sometimes not identify the poles correctly. In those cases, contour points can fall into MreB-msfGFP-free regions at the poles even if the corresponding centerline points are more than just 0.5 um away from the automatically identified poles.

To determine correlations between MreB and contour curvature independently of spontaneous cell bending, we either concentrated on regions where the smoothened centerline was straight or we performed two independent normalization approaches:

To constrain our analysis to straight segments of the cell we considered only those boundary points corresponding to centerline curvatures with $|\kappa_i| < 0.05/\mu\text{m}$.

In the first normalization approach we renormalized the contour curvature by subtracting the contribution of cell bending according to

$$c_i^{\text{corr}} = c_i \pm \kappa_i / \left(1 - \frac{\kappa_i w_i}{2}\right)$$

where $\kappa_i$ is the curvature of the smoothened centerline (x- and y-coordinates Gauss-filtered, $\sigma = 3.5$ steps), and where $w_i$ is the local width of the cell. The centerline was smoothened to consider only the contribution of long-range bending rather than the impact of short-scale oscillations of boundary curvature. Positive/negative centerline-curvature values correspond to cells bent to the right/left along the direction of the centerline. The plus or minus signs correspond to the right or left side of the cell, respectively. The correction only works for segments of the cell with $\kappa_i < 0.5\, w_i$ to avoid divergence. For the cell segments excluding poles and potential septa this criterion was always fulfilled.

In the second normalization approach we renormalized the MreB intensity by the component expected due to cell bending:

$$I_i^{\text{corr}} = I_i / (1 \pm \alpha \kappa_i w_i)$$

where alpha ~ 0.2 is a coefficient that accounts for the correlations observed between MreB intensity and smoothened centerline curvature (**Figure 6—figure supplement 2D**). As above, the plus/minus signs correspond to the inner/outer face of the bent cell, respectively.

## Spot analysis and colocalization analysis on cell boundaries and in TIRF

To analyse the distribution of fluorescence peaks of cells expressing mCherry-PBP2, MreB-msfGFP, or GFP-RodZ and to perform colocalization measurements we first obtained fluorescence profiles on cell contours from epi-fluorescence images as described in 'Curvature analysis and MreB-curvature correlations'.

Peak analysis: Fluorescence profiles were smoothened with a Gauss filter ($\sigma = 0.5$ pixels). Subsequently, we subtracted the median intensity of every cell contour. Peaks were then detected as positive-valued local maxima (**Figure 3C**).

The intensity-dependent peak density is the number of peaks with intensity $p$ divided by total contour length of all cells $\sum_i^{N_{\text{cells}}} l_i$. For the total density of all peaks, we considered all peaks with peak height 3 times higher than the estimated intensity noise (gray regions in **Figure 3C**). The noise was calculated in non-treated conditions. To that end, we filtered the raw fluorescence image with a 2D Gauss filter of $\sigma = 0.5$ pixels and $\sigma = 3$ pixels separately. The signal on the cell boundaries are then extracted from two sets of images. Next, we took the difference of signals belonging to same cells and calculated the average standard deviation of the differences, which is used as a readout for pixel noise.

For colocalization analysis, we extracted profiles as described above for two different fusions and calculate the Pearson correlation coefficient in each cell independently.

For the analysis of spots and colocalization with TIRF microscopy, images of mCherry-PBP2, MreB-msfGFP, MreB-mCherry, and GFP-RodZ are taken with Microscope one in TIRF mode. Cells are manually segmented based on the bright-field image by drawing a line along the long axis of the cell, defining a centerline. The raw florescence images are smoothened using a 2D-Gauss filter ($\sigma$ = 1 pixel). The 2D-intensity map is acquired from the region $\pm$ 210 nm around the centerline. For peak analysis, we subtracted the median intensity from every map and detected peaks as positive-valued local maxima. The intensity-dependent peak density and total density of all peaks are calculated in the same way as on the cell boundary. To calculate the levels of noise, the raw fluorescence images are filtered with a 2D Gauss filter of $\sigma$ = 1 pixel and $\sigma$ = 3 pixels separately. Next, we took the difference of maps belonging to same cells and calculated the average standard deviation of the differences, which is used as a readout for pixel noise.

For colocalization analysis with TIRF microscopy, we first applied a correction for the systematic shifts between the signals appear in the red- and green- channels. By imaging TetraSpeck beads (Invitrogen) in the two channels and performing peak detection, we found a displacement vector $\Delta \boldsymbol{p} = (-1, -0)$ with a magnitude of 105 nm. Then, we modified the coordinates of the points on the centerline of each cell according to: $\boldsymbol{p}_i^c = \boldsymbol{p}_i + \Delta \boldsymbol{p}$ where $\boldsymbol{p}_i$ and $\boldsymbol{p}_i^c$ show the original and modified centerline coordinates, respectively. Next, we generated 2D intensity maps (*Figure 5—figure supplement 3C-D*) and calculated the Pearson correlation coefficient between the two different fluorescence channels for each cell.

## Determination of bound fraction and diffusion constant

We used the Spot-On software (*Hansen et al., 2018*) to fit 2- and 3-state-diffusion models to high-frequency data (*Supplementary file 2*). In brief, Spot-On fits the experimental jump-length distributions for different time lags to the analytical solution of a multi-state diffusion model. The code returns fractions and diffusion constants of the different fractions, and the particle-localization precision. We used jump-length distributions from lags between 1 and 5 time intervals (60–300 ms). To that end, we selected tracks with at least 4 or seven localizations, for PAmCherry and GFP fusions respectively. We used a normalized value for the sum of residuals ($\chi^2$) acquired by dividing $\chi^2$ by the number of bins and by the lag time (in number of time steps) used.

To make sure bound molecules are indeed bound, we checked the value of $D_{\text{bound}}$ by leaving it free for TKL130 and TKL130/pKC128 tracks (*Supplementary file 2b*). $D_{\text{bound}}$ is then found to be between $10^{-4}$-$10^{-9}$ from fitting a 3-state model. After confirming near-zero values for $D_{\text{bound}}$, we fixed $D_{\text{bound}}$ = 0 for all experiments.

We complemented Spot-On measurements by a different method, which is based on the distribution of single-track effective diffusion constants (referred to as '$D_{\text{eff}}$-based method'): For every track, we calculated $D_{\text{eff}}$ from the first 4 steps of each track, applying a linear fit of single-track MSD's $<x^2(t)>$ according to $<x^2(t)>=4D_{\text{eff}}t + 4\sigma^2$. The empirical distribution of $D_{\text{eff}}$ was then compared with distributions generated from computational simulations. For the reference condition of PAmCherry-PBP2 fusions (TKL130), we calculated distributions $p(D_{\text{eff}}|D, \sigma)$ for diffusive molecules with different diffusion constant $D$ and localization uncertainties $\sigma$. The combined distribution of a two-state population is then given by $p(D_{\text{eff}})=b \, p(D_{\text{eff}}|D_{\text{bound}} = 0, \sigma) + (1-b) \, p(D_{\text{eff}}|D, \sigma)$, where we assume bound molecules to be immobile. For every combination $[D, \sigma]$ we obtained the bound fraction $b$ by fitting the peak value at $D_{\text{eff}}$ = 0 $\mu m^2$/s to the experimental data. $[D, \sigma]$ were then obtained by minimizing the residual sum of squares (RSS) between experimental and simulated distributions, considering values for $D_{\text{eff}}$ between $-0.015$ and $0.055$ $\mu m^2$/s (see *Figure 1—figure supplement 3A*). We used a bin size of 0.005 $\mu m^2$/s. We varied $D$ between 0.015 and 0.06 $\mu m^2$/s with an interval of 0.005 $\mu m^2$/s and $\sigma$ between 0 and 40 nm with and interval of 5 nm. The parameter set of $D$ = 0.04 $\mu m^2$/s and $\sigma$ = 20 nm gave the lowest RSS. We confirmed that $D_{\text{bound}}$ = 0 $\mu m^2$/s by comparisons between distributions for zero and finite values of $D_{\text{bound}}$ (*Figure 1—figure supplement 3B*).

We compared the values given by Spot-On with the $D_{\text{eff}}$-based method using a 2-state model and acquired very similar results (*Figure 1—figure supplement 3C*).

## Velocity and orientation distributions of persistently moving PBP2 and MreB

Directed motion of individual trajectories can be inferred from a quadratic dependency of the single-particle mean squared displacement (MSD) on time according to

$$\langle x_i^2 \rangle = v_i^2 t^2 + b_i \tag{1}$$

Here, $v_i$ denotes the velocity of particle $i$ and $b_i$ is an offset reflecting the localization error.

To determine the velocity distribution of persistently moving PBP2 molecules or MreB filaments, respectively, we analyzed a subset of trajectories obtained with 1 s time intervals, for which each single-track MSD could be well fit by a quadratic function of the form of *Equation 1* according to the coefficient of determination $R^2$. We considered trajectories with at least four time steps. For PBP2 we considered trajectories with a maximum of ten time steps to prevent a bias towards slowly moving molecules that are more likely to remain in the TIR field of illumination for long times. MreB trajectories are longer in time than PBP2 trajectories, since they are obtained from filaments containing many msfGFP molecules, and they are obtained from a larger field of view since their movement is measured in epi-fluorescence mode. For comparability with PBP2 tracks we therefore constrained the MreB trajectories by analyzing the 10 displacement steps around the point closest to the cell centerline. These displacements are found within a field of view also detectable in TIRF mode.

The velocity distributions for different minimal $R^2$ values are plotted in *Figure 1—figure supplement 5D*. As expected, low velocities corresponding to immobile molecules contribute less with increasing minimal $R^2$. Accordingly, both MreB and PBP2 showed a mildly increasing average velocity as a function of the minimum $R^2$ (*Figure 1—figure supplement 5D*). For comparison between PBP2 and MreB in *Figure 1F–G* we generated velocity distributions from tracks that showed $R^2$ values > = 0.9, corresponding to ~ 10% of PBP2 tracks that showed the highest $R^2$ values (*Figure 1—figure supplement 5C*), yielding average velocities of 18.3 nm/s for PAmCherry-PBP2 and 16.5 nm/s for MreB-msfGFP filaments (*Figure 1—figure supplement 5D*).

Performing the same analysis on PAmCherry-PBP2 trajectories generated with a time interval of 3.6 s we obtained an average velocity of $v$ = 14.2 nm/s for persistent tracks ($R^2$ > = 0.9). The average velocity is lower than for the 1 s dataset because fast-moving molecules are observed less often than slowly moving molecules, as they leave the TIR field of view at higher probability (*Figure 1—figure supplement 5E-F*). For msfGFP-PBP2 we obtained an average velocity of 13.5 nm/s.

To obtain the orientation of persistent PBP2 tracks we calculated the angle between the end-to-end vector of persistent trajectories measured with time interval of 3.6 s (same criteria of $R^2$ > = 0.9 and a minimum length of 4 time points) and the cell orientation. For MreB we calculated the orientation in the same way and on the same data for which we obtained the velocity distribution.

## Localization accuracy in low-frequency imaging

We determined the experimental localization accuracy from the distribution of displacements from measurements at 3.6 s time intervals for PAmCherry-PBP2 (*Figure 1—figure supplement 6A*). We restricted our analysis to immobile molecules according to a simple criterion (end-to-end distance less than 200 nm for tracks of at least 7 steps). The standard deviation $\sigma_d$ of a Gaussian fit to the distribution of displacements in a single spatial direction is determined by $\sigma_d^2 = \left\langle (x_i - x_{i+1})^2 \right\rangle = 2\sigma^2$, where $\sigma$ is the localization uncertainty. Therefore, $\sigma = \sigma_d/\sqrt{2}$. For PAmCherry-PBP2 $\sigma = 25\,\text{nm}$.

## Simulation of persistent and immobile tracks

To establish a criterion for reliably classifying persistently moving and immobile states in experimental PBP2 trajectories, we computationally simulated tracks of immobile or persistently moving molecules resembling the tracks observed by microscopy using 3.6 s time intervals. To that end we randomly picked a trajectory length (number of steps $n$) from an exponential distribution with $\langle n \rangle = 3.5$ that resembled the experimental length distribution (*Figure 1—figure supplement 6B*). For the simulation of persistently moving or immobile molecules we imposed a constant step size in the x-direction corresponding to the experimental velocity ($v$ = 14 nm/s for PAmCherry-PBP2) for persistent molecules or to $v = 0$ nm for immobile molecules. To account for the localization uncertainty we subsequently added to all x- and y-coordinates a random displacement drawn from a

normal distribution with a mean of 0 nm and a standard deviation equal to the localization accuracy for low-frequency imaging 25 nm (for PAmCherry-PBP2).

Trajectories with transitions between immobile and persistent states were obtained by randomly selecting sub-trajectories from a single 1000-step long trajectory containing transitions between persistent and immobile states. Transition rates were obtained from experimental data (see next paragraph).

## Determination of persistent and immobile states and switching rates

For each time point we calculated the smoothed local velocity by dividing the displacement during the surrounding $w$ time steps by the time lag $w\tau$:

$$v(t) = \boldsymbol{r}\left(t + \frac{w}{2}\right) - \boldsymbol{r}\left(t - \frac{w}{2}\right)/w\tau$$

Here $\boldsymbol{r}(t) = (x(t), y(t))$ is the position at time $t$, and $\tau = 3.6$ s is the imaging time interval. We classified the particle as either immobile or persistently moving at time $t$ if $v$ was smaller or bigger than the threshold velocity $v_{thr}$, respectively.

We applied different values for $w$ and $v_{thr}$ to simulated trajectories of PAmCherry-PBP2 ($v = 14$ nm/s; sigma = 25 nm) to find the parameter combination that reliably detected dynamic states in simulated tracks with more than 99% success rate (*Figure 1—figure supplement 6C*). We chose a window size of $w = 4$ and a velocity threshold of $v_{thr} = 8$ nm/s (classifying 99.6% of segments of simulated immobile molecules as immobile and 99.6% of simulated persistent molecules as persistent) (*Figure 1—figure supplement 6C*). Since we found almost identical average velocity for msfGFP-PBP2, we used the same window size and velocity threshold for persistence classification.

We calculated transition rates $k_{ip}$ or $k_{pi}$ by counting the number of transitions from immobile to persistent or persistent to immobile states, respectively, and dividing by the total duration of persistent or immobile states observed. Here, we ignored intermittent segments of duration of a single time step.

All error bars denote standard errors between replicates.

## Testing the two-state model of immobile and persistent states

To test whether the dynamics of PBP2 molecules is compatible with a model of molecules residing in either of two possible states we measured single-particle MSD's of track segments identified as either immobile or persistent (*Figure 1—figure supplement 6A*). The average MSD of segments classified as persistently moving increased quadratically with time, while the average MSD of segments classified as immobile remained nearly constant (*Figure 1—figure supplement 6D-E*).

To test whether deviations of single-particle MSDs from the average were due to transitions between the two states we analyzed computationally simulated tracks of our two-state model containing transitions. We then used the same classification criterion and MSD analysis on simulated tracks as on experimental tracks. For simulations of persistent segments we adjusted the velocity of $v = 12$ nm/s (compared to 14 nm/s above) that yielded better agreement of the average MSD curves of persistent segments with experiments (*Figure 1—figure supplement 6D*). The difference is likely due to the fact that we consider here all persistent segments while above we obtained the velocity from the 10% most persistent full tracks according to the $R^2$-criterion.

Simulations and experiments showed very similar distributions of single-particle MSD's (*Figure 1—figure supplement 6F*), suggesting that bound PBP2 molecules are indeed either immobile or persistently moving, but not found in a qualitatively different slowly-moving state.

## Calculation of the unbinding rate based on fluorescence-lifetime measurements

To obtain the unbinding rate $k_{bd}$ in non-treated or A22-treated cells we measured lifetime distributions of tracks $f(n, \tau)$ obtained with 1 s exposure time and different time intervals $\tau = 1$ s or 12 s (*Figure 2*). Here, $n$ is the number of steps a track is observed corresponding to the lifetime $t = n\tau$. At first, we assume two random processes to contribute to particle loss: GFP bleaching with a probability $p_b$ per time frame, and a second process with an apparent track termination rate $k_a$, corresponding to a termination probability $p_a = 1 - \exp[-k_a\tau]$.

For A22-treated cells, $k_a$ is caused by unbinding only, that is, $k_{bd} = k_a$. For non-A22-treated cells, $k_a$ subsumes unbinding and particles leaving the field of view due to persistent motion. While the probability of molecules leaving the field of view is not independent of track duration, this assumption does not affect our calculation of $k_{bd}$, as we will see below.

For both conditions (-A22, +A22), we simultaneously fit the two lifetime distributions to the above model, considering tracks between 3 to 7 steps (4–8 localizations). For A22-treated cells we obtained $p_b = 0.43 \pm 0.08$ and $k_a = k_{bd} = 0.021 \pm 0.008$ s$^{-1}$. The unbinding rate corresponds to an average lifetime of the bound state of $48 \pm 18$ s.

For non-A22-treated cells we obtained $p_b = 0.39 \pm 0.08$ and $k_a = 0.035 \pm 0.007$ s$^{-1}$. To estimate the contribution of persistent motion to the apparent unbinding rate, we then conducted simulations of bound molecules transitioning between persistent and immobile states: Molecules started from random positions within the field of view (width 600 nm) either moving persistently with speed of $v = \pm 14$ nm/s perpendicular to the central axis, or resting immobile. Transitions between the two states occurred at experimental rates $k_{ip}$ and $k_{pi}$, respectively. The probability of bleaching was set equal to $p_b$. In a second set of simulations, we changed either of the two rates to maintain the measured persistent fraction of p=80%, that is, either $k_{ip}^{corr} = k_{pi}p/(1-p)$ or $k_{pi}^{corr} = k_{ip}(1-p)/p$ (*Figure 2—figure supplement 1*). We then measured track-length distributions from simulations with different unbinding rates $k_{bd}$ to infer the range of unbinding rates $k_{bd}$ compatible with the experimentally obtained apparent termination rate $k_a$ (*Figure 2—figure supplement 1*).

In both conditions, experimental lifetime distributions of 1 s data are dominated by bleaching. Therefore, the time-dependent process of persistent molecules leaving the field of view only affects the 12 s distributions. While those distributions are not perfectly exponential, we could still fit them by an exponential function over the window of 3 to 7 steps, considered here.

## Estimate of the number of MreB filaments per cell

To estimate the number of MreB filaments we assume for simplicity that all MreB proteins of the cell are part of dimers of MreB protofilaments (*Salje et al., 2011*). A substantial fraction of proteins is found as cytoplasmic monomers. Our estimate is therefore rather conservative. Previous measurements suggest that there is a broad distribution of filament lengths with many filaments as long as 1 um (*Ouzounov et al., 2016*). As a conservative estimate we assume that all filaments are 100 nm in length and have a repeat length of 5 nm (*Salje et al., 2011*). Each of the idealized filaments therefore contains 40 monomers. With an average of 2000 or 11000 proteins per cell in poor or rich growth media, respectively (*Li et al., 2014*) the cytoplasmic membrane is decorated with up to 50–200 filaments according to this simple model.

## Measuring transitions from diffusive to bound states (Bound-Molecule FRAP)

In order to determine the transition rate from the diffusive to the bound state $k_{db}$ we measured the bound fraction at different time points after bleaching the field of view, conceptually similarly to classical FRAP (fluorescence recovery after photobleaching) experiments. In a first step we aimed to activate all PAmCherry fluorophores in the TIRF field of view with a 1.5 s exposure of 10-fold increased UV intensity compared to our standard protocol (see above). We introduced a waiting time of 2 s in the dark in order to let diffusive photo activated PBP2 molecules escape the field of view. Then, we photo bleached for 4 s with normal excitation intensity. After a recovery period of 0–10 min in the dark, we acquired images in high-frequency mode without any additional photo activation for a duration of 48 s. In this way, we were able detect PAmCherry-PBP2 molecules, which were activated in the first step, which were able to escape the field of view during the 2 s pause, and which then reentered the field of view, where they either remained diffusive or bound to their substrate.

For finding bound fractions as reported in *Figure 4B*, we fixed $D_1 = 0.02$ and $D_2 = 0.06$ which are the diffusion constants found in the reference state (A22-treated cells with 20 ug/ml for 30 min) in order to avoid fluctuations in population sizes.

## Estimation of the timescale of molecules leaving the TIR field of view due to persistent motion

To estimate the average time it takes a molecule that bound randomly in the field of view to leave the field of view through persistent motion, we conducted simulations identical to those in 'Calculation of the unbinding rate based on fluorescence-lifetime measurements', ignoring the effect of bleaching. Specifically, bound molecules were initially assigned to be immobile before transitioning between immobile and persistently moving states according to the experimentally determined rates $k_{ip}$ and $k_{pi}$. Persistently moving molecules moved with a the experimentally determined average speed of 14 nm/s in a random direction, while immobile molecules remained at their current locations. According to this model it takes 45 s on average for a molecule to leave the field of view through persistent motion. This time is four times smaller than the time it takes an initially diffusive molecule to bind. We therefore reasoned that the rapid escape of bound molecules through persistent motion is responsible for the FRAP curve of non-A22-treated cells to not show recovery.

## Quantification of expression level from fluorescence

For PAmCherry-PBP2 we counted the number of fluorescent spots observed per cell after a single activation pulse of the UV laser. For the comparison between native and overexpression levels see *Figure 1—figure supplement 1D*.

We quantified the different levels of msfGFP-PBP2 obtained on microscope 2 (see above) by measuring the total GFP fluorescence intensity per cell, where cell outlines were obtained by segmentation of phase-contrast images using Morphometrics (SimTK) (*Ursell et al., 2017*). We subtracted contributions from auto-fluorescence per pixel as obtained from imaging wildtype *E. coli* (MG1655) cells. For a comparison between different induction levels see *Figure 1—figure supplement 8B*.

For comparing the number of bound molecules: We compared PAmCherry-PBP2 track densities between native and overexpression levels. To collecting tracks, we used same activation power for the two strains and acquired five consecutive images with an exposure time of 1 s. Then, we did peak detection and tracking as described in 'PBP2 tracking'. We selected tracks with at least three localizations to discard any mis-annotations. Density of tracks is than found by dividing the number of tracks by the area in the field of view covered with cells (*Figure 1—figure supplement 7C*).

## Calculation of the stoichiometry between RodA and bound PBP2 molecules

For the stoichiometry between RodA and PBP2 in wildtype cells, we used literature values obtained by ribosome profiling (*Li et al., 2014*). In those measurements, the stoichiometry is 1.34 and 1.36 in poor (minimal medium + glucose) and rich growth media (rich defined media), respectively. We thus assume that the mean stoichiometry of 1.35 holds for wildtype cells in our intermediate growth medium (minimal medium+glucose+casamino acids).

In the RodA depletion strain and in the parent PAmCherry-PBP2 overexpression strain, PBP2 levels are about 9-fold higher than in MG1655 according to mass spectrometry, while RodA levels are 1.5-fold above wildtype levels. However, fluorescence measurements suggest that the increase of PAmCherry-PBP2 with respect to TKL130 is only about three-fold, while PAmCherry-PBP2 is expressed between one- to two-fold above wildtype PBP2 (MG1655) in TKL130 depending on the measurement (Bocillin, DIA, PRM). Assuming that functional PAmCherry-PBP2 levels are equal to PBP2 levels in MG1655 as a conservative estimate, we then calculated the stoichiometry of PAmCherry-PBP2 and RodA in the overexpression strain according to $N^0_{RodA}/N = 1.35/3 = 0.45$, where $N_{RodA}$ and $N$ are the numbers of RodA and PBP2 molecules per cell, respectively.

Upon CRISPRi-based repression, we estimate that RodA levels go down as a function of time $t$ according to

$$N_{RodA}(t) = N^0_{RodA}\left[(1-f)2^{-t/t_d} + f\right],$$

where $t_d = 90\,\mathrm{min}$ is the doubling time and $f = 0.1$ is the relative residual expression level of mCherry-PBP2 after CRISPRi-based repression of the *mCherry-mrdA-mrdB* operon during steady-state growth (*Vigouroux et al., 2018*). This strain is equal except for the differences between

PAmCherry and mCherry and for the overexpression of PBP2 from the pKC128 plasmid. We then made the conservative estimate that RodA only started to drop 4h after inducing dCas9 due to any unanticipated delay of dCas9 activity. 5h after induction we observed a significant increase in cell diameter, suggesting that RodA levels had already fallen well below wildtype levels.

Accordingly, we found that $N_{\mathrm{RodA}}/N = 0.1$. With a bound fraction of $b = 0.19 \pm 0.03$, we then obtain the ratio between RodA and bound PBP2 molecules of $N_{\mathrm{RodA}}/(bN) = 0.5 \pm 0.1$. Thus, there is likely less than one RodA molecule for every bound PBP2 molecule.

## Model to test the contribution of diffusing PBP2 molecules to rod-complex activity

Cross-links with neighboring glycan strands are formed every other di-sugar subunit. Each subunit is about 1 nm long (*Boal and Boal, 2012*). Thus, the rate of transpeptidation corresponding to a high but common speed of MreB of 30 nm/s is $\lambda$ = 15/s. Lee et al. argued that after forming one cross link, PBP2 would detach and diffuse in the cell envelope to find a new site for cell-wall cross-linking (*Lee et al., 2014*).

The number of PBP2 enzymes in the cell is about between 100–300 in nutrient-rich medium and 60–75 in poor medium according to radiolabeling (*Dougherty et al., 1996*) or ribosome profiling (*Li et al., 2014*). We thus wondered whether free diffusion of such a small number of enzymes could account for the experimentally observed rate of cross-link formation, or whether free diffusion would limit this process. Alternatively, we also considered that molecules underwent facilitated diffusion along one-dimensional tracks such as the cytoskeleton MreB (*Oswald et al., 2016*), similarly to the phenomenon of transcription factors searching their target on chromosomal DNA (*Mirny et al., 2009*).

We conducted overdamped Brownian-dynamics simulations of N = 100 enzymes [interpolating between measurements made for poor and rich media (*Dougherty et al., 1996*; *Li et al., 2014*) in a rectangular domain of 3 × 3 um with periodic boundary conditions in x- and y-directions, thus approximating the cylindrical surface of a rod-like *E. coli* bacterium of 1 um width and 3 um length (*Figure 1—figure supplement 1B*) and ignoring the shape of the cell poles.

The overdamped Brownian motion of PBP2 in our model is governed by the Langevin equation for its position

$$\dot{\boldsymbol{r}} = D\zeta$$

where the dot denotes a time derivative, D = 0.06 um²/s is the experimental diffusion constant, and $\zeta$ is the zero mean Gaussian white noise random displacement originating from the solvent. Its variance is given by

$$\overline{\zeta(t)\zeta(t')} = 2\delta_{ij}\delta(t-t'), \; i,j = x,y,$$

where the bar denote a noise average.

A number of n = 10 circular cross-linking sites of diameter a = 10 nm are placed at random locations in the rectangular domain. Once a diffusing molecule hits any of the cross-linking sites, a cross-linking event is registered to occur. Note, that this model is based on the conservative estimate that every encounter between enzyme and cross-linking site leads to a successful reaction. To prevent rapid return of an enzyme to the same site we introduce a deterministic latency time $t_{\mathrm{off}}$ after every encounter during which an enzyme can diffuse but not facilitate a reaction. This latency time could reflect the typical time it takes to conduct one reaction or a combination of different microscopic effects. The reaction rate per site $\gamma$ is then calculated as the mean number of enzyme-site encounters per site per total simulated time.

We consider latency times larger than 0.1 ms (*Figure 7B*), a time that is needed for a PBP2 enzyme to explore an area similar to the size of an enzyme (5 nm). In this regime, the encounter rate depends only weakly on $t_{\mathrm{off}}$ (*Figure 7B*). Thus, only a minor fraction of enzymes re-encounters the same site shortly after leaving it (*Figure 7C*). Notably, the effect of $t_{\mathrm{off}}$ on rebinding is much weaker than in the previously studied cases of finding a membrane receptor from the cytoplasm or of binding a receptor in the 3D bulk (*Mugler et al., 2012*), where the probability of rebinding decays algebraically with the latency time for short times and exponentially for long times. Results are nearly independent of target numbers n (not shown).

We next considered the possibility that PBP2 undergoes facilitated diffusion along one-dimensional tracks, such as MreB filaments: MreB forms circumferentially oriented filaments of up to 1 um in length (*Ouzounov et al., 2016*). PBP2 enzymes and other cell-wall proteins interact with MreB filaments (*Kruse et al., 2004*; *Morgenstein et al., 2015*), and PBP2 was observed to partially co-localize with MreB filaments (*Lee et al., 2014*). To test the possible influence of linear tracks on the encounter rate we extended the model introduced above by adding unidirectional filaments of length $l$ to every rod-complex site (filaments are oriented along the $y$-axis). PBP2 molecules cannot cross filaments. Instead, a PBP2 molecules that encounters a filament, diffuses along the filament with the same diffusion constant $D$ until it either a) hits the reaction site, b) reaches one of the two filament ends and returns to 2D diffusion, or c) is randomly displaced from the filament by an amount $\Delta x = 2$ a with rate $k_{off}$. Only after hitting the target is an enzyme inactive for the latency time $t_{off}$.

## Western blotting

In order to estimate the relative amount of msfGFP-PBP2 in different strains or with different induction levels, three independent preparations of the membrane protein fraction of AV127 and TU230 (attLHC943) were analyzed by western blot using a GFP primary antibody.

Cells were grown overnight in LB at 37°C and diluted 1/400 into 15 ml of M63A with varying amounts of IPTG or not depending on the strain. Three independent cultures, for each strain, were grown at 30°C with different inducer concentrations (5 to 25 µM of IPTG) for 6 hr to an OD600 approximately of 0.3. Cells were harvested by centrifugation and resuspended in 500 µl of ice-cold 1X phosphate-buffered saline (PBS) containing 10 mM EDTA. Cells were disrupted by sonication and the membrane protein fraction was collected by centrifugation (20,000 x g for 30 min at 4°C). The membrane fractions were suspended in 100 µL of 1X PBS.

The protein concentrations were determined using a Bradford-based Protein Assay (5000006, Bio-Rad) according to the instructions. Membrane protein fractions were adjusted to the same concentration with 1X PBS. 75 µl of the membrane protein extract was mixed with 25 µl of Laemmli sample buffer 4X (#1610747, Bio-Rad). Ultimately, approx. 14 µg of proteins for each sample were loaded and separated on 10% SDS-PAGE gels (Miniprotean TGX, Bio-rad). After migration, the proteins were transferred on PVDF membranes. The membranes were incubated for 1 hr in TRIS-buffered saline, 0,1% Tween 20 (TBS-T) with 3% milk at room temperature and incubated in TBS-T milk 3% with anti-GFP antibodies (1/10000 dilution) ON at 4°C. Membranes were then washed three times with TBS-T and incubated for 1 hr with the secondary antibody tagged with horseradish peroxidase (HRP) (Goat anti-rabbit #172 1019, Bio-Rad) at room temperature. Unbound secondary antibodies were again washed out with three TBS-T washes. Signal was revealed using ECL solution (RPN2232, Amersham) and blots were imaged. Relative fold-change in signal intensity was measured in ImageJ.

## PBP2 Mass-spectrometry

To quantify relative changes of protein levels between conditions and strains, we used Data Independent Acquisitions (DIA) following *Bruderer et al. (2017)*. For absolute quantification of PBP2 levels, we used a targeted proteomics approach by Parallel Reaction Monitoring (PRM) (*Bourmaud et al., 2016*; *Gallien et al., 2012*; *Peterson et al., 2012*).

## Preparation of *E. coli* whole protein extracts

Cells were collected by centrifugation (4000 g, 10 min at 4°C) around $OD_{600}$ 0.15. For absolute quantification of PBP2, an aliquot part was taken from each culture in order to determine cell number by colony counting. Supernatant was removed and cell pellets were flash-freeze in liquid nitrogen and stored at −80°C. Cells were suspended in 250 µl of Urea buffer 8M (Sigma U4883). Cooled cells were lysed by sonication (Fisherbrand FB120) (alternating 3 cycles of 30 s ON with 40% amplitude and 15 s OFF to cool down the sample). Protein concentration was determined using a Bradford-based colorimetric assay (Bio-Rad 5000006; *Bradford, 1976*) with known concentrations of bovine serum albumin (Sigma) as a standard. Proteins samples were diluted with 2x phosphate buffered saline (PBS) in order to decrease Urea concentration and be compatible with the colorimetric assay.

For the quantification of absolute numbers of PBP2 we used colony counting and measured protein concentration led, which resulted in an average of 105 fg of proteins per cell.

## Digestion of proteins

All protein samples were denatured in 8 M urea in Tris HCl 100 mM pH 8.0. Proteins disulfide bonds were reduced with 5 mM tris (2-carboxyethyl)phosphine (TCEP) for 20 min at 23° C and further alkylated with 20 mM iodoacetamide for 30 min at room temperature in the dark. Subsequently, LysC (Promega) was added for the first digestion step (protein to Lys-C ratio = 80:1) for 3 hr at 30° C. Then the sample was diluted to 1 M urea with 100 mM Tris pH 8.0, and trypsin (Promega) was added to the sample at a ratio of 50:1(w/w) of protein to enzyme for 8 hr at 37° C. Proteolysis was stopped by adding 1% formic acid (FA). Resulting peptides were desalted using Sep-Pak SPE cartridge (Waters) according to manufacturer instructions. Peptides elution was done using a 50% acetonitrile (ACN), 0.1% FA buffer. Eluted peptides were lyophilized and then stored until use.

For Data Independent Acquisitions (DIA) and Parallel Reaction Monitoring (PRM) (see below), iRT peptides (Biognosys) were spiked into all samples as recommended by manufacturer.

## Peptide fractionation for spectral library

Peptide fractionation was done using poly(styrenedivinylbenzene) reverse phase sulfonate (SDB-RPS) stage-tips method as described in *Kulak et al. (2014)*; *Rappsilber et al. (2007)*. Briefly, 3 SDB-RPS Empore discs were stacked on a P200 tip and used to fractionate 30 μg of peptides. Four serial elutions were applied as following: elution 1 (80mM Ammonium formate, 20% (v/v) ACN, 0.5% (v/v) FA), elution 2 (110mM Ammonium formate, 35% (v/v) ACN, 0.5% (v/v) FA), elution 3 (150mM Ammonium formate, 50% (v/v) ACN, 0.5% (v/v) FA) and elution 4 (80% (v/v) ACN, 5% (v/v) ammonium hydroxide).

All fractions were dried and resuspended in 0.1% formic acid before injection. For all fractions, iRT peptides were spiked as recommended by Biognosys.

## LC-MS data acquisitions

### Data Independent Acquisitions (DIA)

LC-MS/SM analysis of digested peptides was performed on an Orbitrap Q Exactive HF mass spectrometer (Thermo Fisher Scientific, Bremen) coupled to an EASY-nLC 1200 (Thermo Fisher Scientific). Peptides were loaded and separated at 250 nl/min on a home-made C18 50 cm capillary column picotip silica emitter tip (75 μm diameter filled with 1.9 μm Reprosil-Pur Basic C18-HD resin, (Dr. Maisch GmbH, Ammerbuch-Entringen, Germany)) equilibrated in solvent A (2% ACN, 0.1% FA). Peptides were eluted using a gradient of solvent B (80% ACN, 0.1% FA) from 3% to 6% in 5 min, 6% to 29% in 130 min, 29% to 56% in 26 min, 56% to 90% in 5 min (total length of the chromatographic run was 180 min including high ACN level steps and column regeneration). Mass spectra were acquired in data-independent acquisition mode with the XCalibur 4.1.31.9 software (Thermo Fisher Scientific, Bremen).

Each cycle was built up as follows: one full MS scan at resolution 30 000 (scan range between 400 and 1200 m/z), AGC was set at $3*10^6$, ion trap was set at 50 ms. All MS1 was followed by 40 isolation windows of 20 m/z, covering the MS1 range from 400 m/z to 1200 m/z. The AGC target was $2*10^5$, and NCE was set to 27. All acquisitions were done in positive and profile mode.

### Parallel Reaction Monitoring acquisitions for absolute quantification of PBP2 (PRM)

Peptides chosen and used for absolute quantification of PBP2 were based on the FASTA sequence obtained from UniprotKB database (*UniProt Consortium, 2015*) and MS evidence of identification. Peptides sequences are SGTAQVFGLK and VDNVQQTLDALR (Aqua UltimateHeavy, Thermo Fisher Scientific). Targeted peptides and their heavy forms were imported into Skyline (*MacLean et al., 2010*) to generate precursor ion inclusion list that also contained instrument control parameters for Xcalibur to detect peptides using PRM-MS. Information on iRT peptides (Biognosys) were also generated.

Heavy peptides synthetized from PBP2 sequence were spiked at 16 fmol.μl$^{-1}$ in each sample. Each sample was injected at a known concentration with iRT peptides (as recommended by

Biognosys) and 50 fmol of heavy peptides. Quantity of peptide injected on column was controlled by UV absorbance at 280 nm and tryptophan absorbance.

PRM was performed on an Orbitrap Q Exactive HF mass spectrometer (Thermo Fisher Scientific, Bremen) coupled to an EASY-nLC 1200 (Thermo Fisher Scientific). Peptides were loaded and separated at 250 nl.min$^{-1}$ on a home-made C18 50 cm capillary column picotip silica emitter tip (75 μm diameter filled with 1.9 μm Reprosil-Pur Basic C18-HD resin, (Dr. Maisch GmbH, Ammerbuch-Entringen, Germany) equilibrated in solvent A (2% ACN, 0.1% FA). Peptides were eluted using a gradient of solvent B (80% ACN, 0.1% FA) from 5% to 10% in 1 min, 10% to 30% in 82 min, 30% to 50% in 5 min, 50% to 95% in 5 min (total length of the chromatographic run was 105 min including high ACN level steps and column regeneration). Mass spectra were acquired XCalibur 4.1.31.9 software (Thermo Fisher Scientific, Bremen). The acquisition method combined a full scan method with a time-scheduled sequential PRM method. For the full MS, a scan range of 350 to 1500 m/z, an orbitrap resolution of 60000, and an AGC value of a $3*10^6$ were used. An orbitrap resolution of 60000, a maximum IT set at 110 ms, an isolation window selection of 1.2 m/z, AGC target was $2*10^5$ and NCE fixed at 28 were used. Targeted, heavy and retention time peptides (iRT peptides, Biognosys) were listed in an inclusion list and monitored.

## Data analysis

### Data analysis for spectrum library building and DDA analysis of Co-IP

For spectral library purposes and DDA experiments, MaxQuant (*Tyanova et al., 2016a*) 1.5.5.3 was used. Raw data were analyzed against an *E. coli* database (6071 entries, downloaded from Uniprot on 10/03/2016).

The following search parameters were applied: carbamidomethylation of cysteines was set as a fixed modification, oxidation of methionine and protein N-terminal acetylation were set as variable modifications. The mass tolerances in MS and MS/MS were set to 5 ppm and 20 ppm respectively. Maximum peptide charge was set to 7 and 7 amino acids were required as minimum peptide length. A false discovery rate of 1% was set up for both protein and peptide levels.

Data analysis was done mainly using Excel and Perseus environment (*Tyanova et al., 2016b*).

### Data analysis for DIA acquisitions

DIA experiments were analyzed using Spectronaut X (v. 11 Biognosys AG). Dynamic mass tolerance at the MS1 and MS2 levels was employed. The XIC RT Extraction Window was set to Dynamic with a correction factor of 1. Calibration mode was set to automatic with nonlinear iRT calibration and precision iRT enabled. Decoys were generated using the scrambled method and a dynamic limit (default settings). P value estimation was performed using a kernel density estimator. Interference correction was enabled with no proteotypicity filter. Major grouping was by Protein-Group ID, and minor grouping was by stripped sequence. The major group quantity was mean peptide quantity. The major group top N was enabled with a minimum of 1 and a maximum of 3. Minor group quantity was mean precursor quantity. The minor group top N was enabled with a minimum of 1 and a maximum of 3. The quantity MS-Level was MS2, and quantity type was area. Q value was used for data filtering. Cross run normalization was enabled with Q value sparse row selection and local normalization. The default labeling type was label-free with no profiling strategy and unify peptide peaks not enabled. The protein inference workflow was set to automatic.

### PRM data analysis

Raw mass spectrometry data were exported to Skyline-daily (version 4.1.1.18179) for identification of transitions and peak area integration. Data were exported in. csv file format and analyzed in Excel.

## Acknowledgements

We thank Timothy Lee and KC Huang for strain TKL130 and plasmid pKC128, Nikolay Ouzounov and Zemer Gitai for strains NO50 and NO53, Tom Bernhardt for strains TU230(attLHC943) and TU230(attLPR122), Felipe Bendezú and Piet de Boer for plasmids pFB121, pFB290 and strain FB60 (iFB273). This work was supported by the European Research Council (ERC) under the Europe

Union's Horizon 2020 research and innovation program [Grant Agreement No. (679980)], the French Government's Investissement d'Avenir program Laboratoire d'Excellence 'Integrative Biology of Emerging Infectious Diseases' (ANR-10-LABX-62-IBEID), the Mairie de Paris 'Emergence(s)' program, the PRESTIGE Postdoc fellowship (Campus France), and the Volkswagen Foundation.

## Additional information

### Funding

| Funder | Grant reference number | Author |
|---|---|---|
| H2020 European Research Council | 679980 | Sven van Teeffelen |
| Agence Nationale de la Recherche | ANR-10-LABX-62-IBEID | Sven van Teeffelen |
| Volkswagen Foundation | | Sven van Teeffelen |
| Mairie de Paris | Emergence(s) program | Sven van Teeffelen |
| Campus France | Prestige Postdoctoral Fellowship | Eva Wollrab |

The funders had no role in study design, data collection and interpretation, or the decision to submit the work for publication.

### Author contributions
Gizem Özbaykal, Conceptualization, Data curation, Software, Formal analysis, Validation, Investigation, Visualization, Methodology; Eva Wollrab, Conceptualization, Data curation, Software, Formal analysis, Validation, Investigation, Methodology; Francois Simon, Data curation, Formal analysis, Investigation, Methodology; Antoine Vigouroux, Resources, Investigation; Baptiste Cordier, Data curation, Formal analysis, Investigation; Andrey Aristov, Software, Validation, Methodology, Andrey implemented and validated the Spot-On analysis of high-frequency movies; Thibault Chaze, Mariette Matondo, Data curation, Formal analysis, Validation, Investigation, Methodology; Sven van Teeffelen, Conceptualization, Data curation, Software, Formal analysis, Supervision, Funding acquisition, Validation, Investigation, Visualization, Methodology, Project administration

### Author ORCIDs
Gizem Özbaykal (iD) https://orcid.org/0000-0002-9115-7120
Antoine Vigouroux (iD) http://orcid.org/0000-0002-8398-5073
Sven van Teeffelen (iD) https://orcid.org/0000-0002-0877-1294

### Decision letter and Author response
Decision letter https://doi.org/10.7554/eLife.50629.sa1
Author response https://doi.org/10.7554/eLife.50629.sa2

## Additional files

### Supplementary files
• Source code 1. Custom written MATLAB code used for image analysis.

• Source data 1. Source data, specifically raw tracks, are provided (one file with x-, y- coordinates and track identifier per replicate).

• Supplementary file 1. PBP2 levels and dynamics. (a) Unperturbed conditions. Measurement results at different levels of PAmCherry-PBP2 and msfGFP-PBP2 as well as msfGFP-PBP2(L61R). Protein levels are measured based on fluorescence, mass spectrometry (DIA and PRM), and Western Blots. (b) PBP2 dynamics under different perturbations corresponding to *Figures 3* and *5*.

- Supplementary file 2. Comparison of 2- and 3-state diffusion models. (a, b) Measurements of PAm-Cherry-PBP2 in TKL130 and TKL130/pKC128, constraining $D_0 = 0$ (a) or leaving $D_0$ free (b). (c) All measurements carried out with the msfGFP-PBP2 or msfGFP-PBP2(L61R) fusions.
- Supplementary file 3. Strains (a), plasmids (b), and primers (c) used in this study.
- Transparent reporting form

### Data availability

All data generated or analysed during this study are included in supplemental datasets provided for each figure. Source data, specifically raw tracks, are provided as Source Data 1 (one file with x-, y-coordinates and track identifier per replicate).

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
