## [Decision Letter]

**Acceptance summary:**

This article provides high-quality measurements that challenge an existing model proposing that cell wall synthesis is spatially controlled by MreB polymers assembling in negatively curved regions of the cytoplasmic membrane. Instead, the positional cue appears to be located on the peptidoglycan side, signaled by the recruitment of the major transpeptidase PBP2, which then provokes assembly of the cell wall synthesis machinery. In the future, it will be important to determine the exact spatial landmark that initiates this cascade.

**Decision letter after peer review:**

Thank you for submitting your article "Transpeptidase PBP2 governs initial localization and activity of major cell-wall synthesis machinery in *Escherichia coli*" for consideration by *eLife*. Your article has been reviewed by two peer reviewers, and the evaluation has been overseen by a Reviewing Editor and Gisela Storz as the Senior Editor. The reviewers have opted to remain anonymous.

The reviewers have discussed the reviews with one another and the Reviewing Editor has drafted this decision to help you prepare a revised submission.

In this report the authors use different fluorescent fusions to track PBP2 localization. They find two main states: diffusive and bound (immobile) when imaging at high frequency. However, lower frequency imaging revealed that the bound fraction consists of both immobile and persistently moving particles, with the persistently moving particles traveling circumferentially as observed previously. The key observation is that the fraction of bound PBP2 particles remains constant over a roughly 3-6 fold range of PBP2 protein levels. This finding indicates that PBP2 is likely to be limiting for the formation of active Rod complexes. In characterizing this motion, the authors present evidence that other components of the rod machinery are not necessary for PBP2 to adopt its bound state, suggesting it is recognizing a partner outside of the complex, which very likely may be the cell wall itself. Importantly, a series of papers have been published that highlight the preference of MreB to localize at sites of negative cell curvature, leading to a model where MreB brings the machinery to these sites to correct aberrant curvature and maintain rod shape. The authors also present data that the prior analysis of MreB curvature preference was dominated by the curvature of the polar regions. When these cellular regions are excluded, the preference is minimal and can be explained by MreB persistent motion. In total, the authors present a convincing case that PBP2 is limiting for Rod system activation, thus nicely complementing the recent report of an activated PBP2 mutant that stimulates Rod system activity in vivo and in vitro. These findings are changing the paradigm of Rod system regulation and will be of interest to a broad audience.

Major changes that must be made to the manuscript before it can be accepted:

1) The data are rigorously collected, assembled and analysed, and the findings are in general appropriately discussed in the context of the data obtained and of the existing literature. However, the paper is currently not written for a sufficiently broad audience. The text is too long, not very well structured and sometimes not an easy read. The figures could be more concise and require the reader to continuously go back and forward. Rewriting in a more concise manner (without sacrificing information), would significantly streamline the paper and convey the logic and connections more efficiently. Ways to clarify confusing presentation of the data are further proposed below.

2) With the present data, the authors cannot strongly conclude that the formation of the bound PBP2 state is independent of Rod system components other than MreB. The data for MreB are convincing due to the use of A22 and its known ability from Park's labeling studies to block PG synthesis by the Rod complex. However, the dependence of the PBP2 bound state on the other components was based on the use of depletion strains. It is therefore entirely possible that the residual bound and persistently moving states observed following depletion reflect the activity of remaining molecules of the depleted proteins. This possibility is especially true for the MreCD and RodZ experiments since they were produced and depleted from multicopy plasmids. The magnitude of depletion observed is significant, but how the final level of protein compares to the native levels is not presented. Also, even if the depletion is significant relative to native levels, it may be that very small amounts of these proteins are sufficient. In absence of additional controls, this section of the results and the associated discussion (and comments in the Introduction) requires significant revision to soften the conclusions and address the possibility that residual Rod components may underly some of the observed phenomena. In fact, reworking this section will help address the first requirement, because the text of this Results section is quite confusing. Also, Figure 3 and 6 might be somehow combined for them to be compared easily, the reader needs to refer to each back and forth (and the multiple supplementary figures) to follow the text.

3) The explanation given to the FRAP experiment of A22-untreated cells is not satisfying. The authors see no recovery of the bound fraction and indicate that this is presumably because bound molecules left the field of view through persistent motion. It is not clear why this would prevent diffusive molecules to transit to the bound state (immobile included) if the binding sites are not saturated and the average lifetime of a bound molecule is about 1 min.

4) Subsection “PBP2 determines the location of new Rod complexes”: In addition to the two possibilities considered by the authors, there is potentially a third possibility: an MreB filament could be present at the PBP2-binding site prior to PBP2 but this filament would not be required for PBP2 recruitment. This could happen if PBP2 and MreB get there independently of each other, recruited by, for ex a third protein. In absence of data showing that PBP2 determines MreB localization this cannot be completely excluded.

Reviewer #1:

This very nice paper from Wollrab and co-workers describes an imaging analysis aimed at understanding how the cell elongation system of *E. coli* is controlled. This system inserts new peptidoglycan (PG) along the cell cylinder and is often called the Rod system or elongasome. Components of the complex include MreB, RodA, PBP2, MreC, MreD, and RodZ. Fluorescent fusions to members of this complex have previously been shown to form foci that rotate around the cell cylinder in a circumferential motion that depends on cell wall synthesis. Therefore, this persistent motion is thought to reflect the active machinery working to insert new PG into the existing cell wall matrix.

Prior work investigating the factors that control Rod system activation has focused on the actin-like MreB protein. A series of papers have been published that highlight the preference of MreB to localize at sites of negative cell curvature, leading to a model where MreB brings the machinery to these sites to correct aberrant curvature and maintain rod shape. This paper, along with a recently published genetic analysis of Rod system function, strongly suggest that rather than MreB, it is the crosslinking enzyme PBP2 that plays the major role in governing Rod system activity.

In this report the authors use different fluorescent fusions to track PBP2 localization. They find two main states: diffusive and bound (immobile) when imaging at high frequency. However, lower frequency imaging revealed that the bound fraction consists of both immobile and persistently moving particles, with the persistently moving particles traveling circumferentially as observed previously. The key observation is that the fraction of bound PBP2 particles remains constant over a roughly 3-6 fold range of PBP2 protein levels. This finding indicates that PBP2 is likely to be limiting for the formation of active Rod complexes. In characterizing this motion, the authors present evidence that other components of the rod machinery are not necessary for PBP2 to adopt its bound state, suggesting it is recognizing a partner outside of the complex, which very likely may be the cell wall itself. Importantly, the authors also present data that the prior analysis of MreB curvature preference was dominated by the curvature of the polar regions. When these cellular regions are excluded, the preference is minimal and can be explained by MreB persistent motion. In total, the authors present a convincing case that PBP2 is limiting for Rod system activation, thus nicely complementing the recent report of an activated PBP2 mutant that stimulates Rod system activity in vivo and in vitro. These findings are changing the paradigm of Rod system regulation and will be of interest to a broad audience.

The paper is clearly written and the data well presented. The experiments and analyses also appear well done, although I am not an expert in single particle tracking and will therefore leave commentary on this aspect of the paper to other more expert reviewers.

I have one major disagreement with the conclusions made by the authors that I think could largely be handled with changes to the text.

Major point:

1) I do not think the authors can strongly conclude that the formation of the bound PBP2 state is independent of Rod system components other than MreB. The data for MreB are convincing due to the use of A22 and its known ability from Park's labeling studies to block PG synthesis by the Rod complex. However, the dependence of the PBP2 bound state on the other components was based on the use of depletion strains. It is therefore entirely possible that the residual bound and persistently moving states observed following depletion reflect the activity of remaining molecules of the depleted proteins. This possibility is especially true for the MreCD and RodZ experiments since they were produced and depleted from multicopy plasmids. The magnitude of depletion observed is significant, but how the final level of protein compares to the native levels is not presented. Also, even if the depletion is significant relative to native levels, it may be that very small amounts of these proteins are sufficient.

Based on ribosomal profiling, *E. coli* cells in defined complete medium are predicted to produce 324 molecules of PBP2. Thus, in the minimal medium used for the microscopy experiments described, it is safe to assume roughly 200-400 PBP2 molecules/cell. According to the data in this report, only 20% of these are bound, and of these only 40% are persistently moving. This would equate to roughly 20-40 active PBP2 molecules/cell at any given time. With the exception of MreB, all the other Rod system components are predicted to be produced at the same level (RodA and MreD), or at twice (MreC) or 3-4x the level of PBP2 (RodZ). If it is true that there are only 20-40 active PBP2 molecules/cell, one can imagine a scenario where the other Rod components could be depleted to levels at or below detection limits, yet still promote the formation of the bound states of PBP2 and play a required role in the activation of the Rod complex. This scenario seems much more likely to me than postulating that WT PBP2 works without the other Rod components and that it may even partner with other PG polymerases to maintain persistent motion upon depletion of RodA. This section of the results and the associated discussion (and comments in the Introduction) requires significant revision to soften the conclusions and address the possibility that residual Rod components may underly some of the observed phenomena.

Reviewer #2:

This manuscript from the van Teeffelen lab is a potentially important contribution to the cell wall biogenesis field. Using single molecule localization and computational simulations, the authors quantify the dynamic behaviours of the essential PBP2 transpeptidase and revisit the prevailing model according to which recruitment of MreB filaments to sites of negative curvature in the membrane initiate Rod complex formation in E coli. Based on their findings, the authors conclude that such model is incorrect and that, instead, Rod complex formation is initiated upon binding of PBP2 to an unknown substrate that is different to MreB and probably a local feature of the cell wall (although no real experimental evidence is provided to support this).

Overall the paper is well presented although the text is too long, not very well structured and quite indigestible to read, and the figures (too many, with too much data that are not always presented in a concise way and require the reader to continuously go back and forward) are also overloaded in my opinion. The data were nevertheless rigorously collected, assembled and analysed, and findings are in general appropriately discussed in the context of the data obtained and of the existing literature. I am not a physicist and thus will not review/comment on the methodological details behind their single molecule tracking/image analysis and simulations (which are about 4/5th of the Materials and methods) – I leave that to the specialists and will focus my comments in the biological part, data interpretation and impact.

I believe that this is an interesting study for the cell wall morphogenesis field. For this work to be highly influential research some mechanistic insight or experimental evidence about the substrate to which PBP2 binds to should be provided (it is speculated that it is some local cell wall feature but no data are provided to directly support this claim), and/or showing that initial localization of the Rod complex depends on PBP2 binding to such peptidoglycan cue and not on other proteins acting upstream PBP2. Along these lines, I am not completely convinced about the RodZ depletion and RodZ-PBP2 co-localization experiments presented and the conclusions that the authors draw from them and I would need them to address this point. This said, the data presented are in general very solid and the conclusions and discussions sound, and to my knowledge accurate quantification of the transitions between dynamic states in single trajectories was not done previously in the field. All in all, I believe that this study is significant enough for publication in *eLife*, and that no significant new work is required if the authors answer to the main points raised by the reviewers. The manuscript requires, however, some work – figures and text should be simplified/shorten for clarity, data of the wild-type strain should be included in the figures for reference, and some parts included in the results could be moved to the discussion.

Most substantial comments

– The explanation given to the FRAP experiment of A22-untreated cells is not satisfying. The authors see no recovery of the bound fraction and indicate that this is presumably because bound molecules left the field of view through persistent motion. I do not see why this would prevent diffusive molecules to transit to the bound state (immobile included) if the binding sites are not saturated and the average lifetime of a bound molecule is about 1 min. Could the authors please comment on this and maybe clarify their explanation in the text?

– Subsection “PBP2 determines the location of new Rod complexes”: In addition to the two possibilities considered by the authors, couldn't an MreB filament be present at the PBP2-binding site prior to PBP2 but not required for its recruitment? Could PBP2 and MreB get there independently of each other, recruited by, for ex a third protein? Why not? Why only the two possibilities indicated? Unless I missed something here or it is conclusively shown that MreB localization is dependent on PBP2 their hypothesis are incomplete and their conclusion biased.

– The entire depletion section should be reworked for the sake of clarity. There is too much information in the figures and often not described in the text. The depletion strains display levels (of MreCD in particular but also of RodZ) well above the native levels at the time of “induction” to start with. Since the bound fraction is increased in the rodZ induced condition and decreased in the RodZ depletion, this suggest that RodZ may recruit PBP2. The authors then exclude this possibility by showing that upon a longer depletion the bound fraction increases again. Yet they need to show that depletion is still working.g – only t6 and t9 are shown. And RodZ levels seem higher at 9h than at 6h (and data are probably from different cells relative to the data shown in Figure 5 supplements). RodZ levels at all the depletion points shown in Figure 5—S4 need to be shown to demonstrate that the depletion is still operating and progressing after 2h and to thus to exclude RodZ as binding cue. Also, the colocalisation experiments RodZ-PBP2 do not add much the way they are presented. Why weren't these done by TIRF? In any case, there is sufficient overlay for a RodZ-dependent binding not to be completely excluded. Overall the text of this Results section is quite confusing and does not convincingly support the strong conclusion of the authors that PBP2 substrate is the cell wall. Also, Figure 3 and 6 might be somehow combined for them to be compared easily, the reader needs to jungle between the two (and the multiple supplementary figures) to follow the text.

[Editors' note: further revisions were suggested prior to acceptance, as described below.]

Thank you for resubmitting your work entitled "Transpeptidase PBP2 governs initial localization and activity of major cell-wall synthesis machinery in *Escherichia coli*" for further consideration by *eLife*. Your revised article has been evaluated by Gisela Storz (Senior Editor), a Reviewing Editor (Tâm Mignot) and one of the previous reviewers.

The manuscript has been improved but there are some remaining issues that need to be addressed before acceptance, as outlined below:

The authors have gone through a large number of experiments, analysis and revisions that make this a better paper. They have done a very good job in addressing most of the reviewers comments. Additionally, reinvestigation of new and old high frequency acquisition data of PBP2 revealed that a 3-state diffusion model (instead of a 2-state diffusion model) was a better fit for their dataset, so they reanalysed their data using a published tool instead of the custom approach (of fitting to a 2-state model) used in the previous version of the manuscript. While the outcome of the 2- and 3- state model were similar for wild-type levels of PBP2 fusions, the outcome for the overexpression of PBP2 was different: the fraction of bound PBP2 particles no longer remained constant (i.e. the number of bound PBP2 increased in direct proportion to total PBP2 levels) but decreased when the levels of PBP2 increased. Thus, the new fitting and analysis affected one of the main messages of the paper: the authors previously claimed that PBP2 was the limiting factor for rod-complex activity and the new results indicate that this is not the case. PBP2 is still important (since the number of active PBP2 molecules still rises, although not in direct proportion, with PBP2 levels) but is not the only limiting factor for Rod activity. The authors have adjusted their manuscript accordingly and provide the comparison of the 2- and 3- state model in their rebuttal letter and in Figure 1—figure supplement 1. I salute them for their in depth analysis and for revisiting their data and conclusions accordingly.

The paper is now clearer (the authors made an effort to shorten the text; discussion is also more structured and sharper) and convincing. The authors have made a clear effort to tone down their claims and indicate that a number of key questions remain to be addressed. The supplementary figures are helpful and improve the manuscript.

Some finals points that need to be addressed prior to publication are:

1) No evidence is provided to make the claim that the cell wall itself is the partner that PBP2 is recognizing outside the Rod complex, other than the slow diffusion coefficient of PBP2. The authors should either give more direct evidence that supports this or else rephrase their sentences to clearly indicate that this is speculation. The authors show that MreB is not the binding substrate of PBP2, and that none of the other Rod components provides the sole substrate for PBP2 binding… but no data are provided to directly support that a local cell wall feature binds PBP2 and initiates Rod complex formation. The authors should tone down this conclusion in the Abstract and as a main conclusion from the results reported.

2) The arguments given to exclude RodZ as PBP2 recruiting factor are not fully convincing on the basis of the data presented. The fraction of bound PBP2 decreases sharply and then progressively increases again (but remains lower than wt levels) during RodZ depletion (from t8). As pointed out during the first round of review, the authors do not show if depletion of RodZ is still operating at later times (the levels of RodZ in the depletion strain are not shown, the text suggests that this is shown in Figure 5—figure supplement 2F but this is not the case). The dCas9 data only shows t3 and t6, while the increase of bound PBP2 fraction operates after t8 in their depletion strain. This point is therefore not fully demonstrated and the conclusion should be toned-down.

3) The explanation that the diffusion constant of the diffusive population of PBP2 decreases significantly upon PBP2 overexpression (“perturbations in the stoichiometry.…. might lead to alternations of the cell wall structure”) is also pure speculation.

4) Title: please correct "Tranpeptidase PBP2 governs initial localization and activity of major cell-wall synthesis machinery in *Escherichia coli*" to"THE transpeptidase PBP2 governs initial localization and activity of THE major cell-wall synthesis machinery in *Escherichia coli*"?

---

## [Author Response]

In this report the authors use different fluorescent fusions to track PBP2 localization. […] These findings are changing the paradigm of Rod system regulation and will be of interest to a broad audience.

We thank the reviewers and the reviewing editor for their appreciation of our work and have addressed all of their concerns and questions as detailed below.

Before answering the specific requests, we would like to point out a finding that affects one of the messages of the paper, namely the limiting role of PBP2 for rod-complex activity: After re-investigating previous and newly acquired high-frequency data of both PAmCherry- and msfGFP-PBP2 fusions we realized that datasets from PBP2-overexpression were better fit by a 3-state diffusion model than by a 2-state diffusion model, i.e., by a model that contains a bound and two (instead of one) diffusive populations: a slowly-diffusing population (*D*_1_ = 0.008-0.015 um^2^/s) and a fast-diffusing population (*D*_2_ = 0.03-0.06 um^2^/s). To that end we switched from our custom approach (of fitting the distribution of apparent diffusion coefficients to a 2-state model) to the published and wide-spread Spot-On tool (Hansen et al., 2018), which is based on fitting multiple distributions of single-molecule-jump lengths (see the new Figure 1B and the new Figure 1—figure supplement 2).

While results of 2- and 3-state model are similar for near-wildtype levels of PAmCherry-PBP2 and msfGFP-PBP2, the 3-state model predicts that the bound fraction decreases at higher expression level (new Figure 1—figure supplement 2, 8). Here, we show in Author response image 1 the comparison between 2- and 3-state models for two different expression levels using PAmCherry-PBP2 (A) and msfGFP-PBP2 (B).

**Author response image 1. respfig1:** Bound fractions and average diffusion constants of the diffusive fraction(s). (**A**) Left: TKL130 (near-native levels). Right: TKL130/pKC128 (overexpression). (**B**) Left: AV127 (*mrdA::sfgfp-mrdA*). Right: inducible msfGFP-PBP2 (*mrdA::aph* (P_lac_::*msfgfp-mrdA*)).

To measure the dynamics of the msfGFP-PBP2 fusion at native levels we now constructed a strain that expresses this fusion from the native *mrdA* locus (AV127) – see also the more detailed explanation further below. For better comparability between msfGFP-PBP2-expressing strains we now repeated all previous measurements of the bound fractions (also including the PBP2(L61R) mutant, and A22-treatment, now conducted in AV127) (Figure 1—figure supplement 8,9; Figure 3—figure supplement 3). In Figure 1—figure supplement 8C, we plot the bound fraction for native and mutant (L61R) msfGFP fusions using the 3-state model.

Due to the decrease of the bound fraction with increasing protein expression, the number of bound and active PBP2 increases less strongly with PBP2 expression level than previously thought. We now calculated the number of bound and active PBP2 molecules and show them as a function of expression level, respectively in a new Figure 1—figure supplement 7.

Depending on the measurement method of PBP2 protein levels (mass spectrometry or Western Blot or fluorescence level), estimates of protein levels and bound/active molecules differ. However, independently of the method, we observed that the number of active PBP2 rises with expression level once PBP2 is overexpressed beyond native levels. This is particularly robust for the PAmCherry fusion, where the bound fraction is measured with high accuracy, due to the lack of background fluorescence in high-frequency videos. We now also confirmed this increase of bound molecules by measuring the density of tracks obtained by low-frequency imaging (panel C). Therefore, PBP2 is still an important limiting factor for Rod-complex activity. However, since neither the number of bound (left) nor active (right) molecules rises in direct proportion to PBP2 levels, PBP2 is not the only limiting factor for activity.

We analyzed all high-frequency data (presented in Figures 1, 3, 4, 5 and supplementary figures) with Spot-On, fitting data to both 2-state and 3-state diffusion models (data presented in Supplementary file 2) and presented the parameters of the 3-state model in the manuscript text.

Where the 2-state model fit the data well (in most cases, where PBP2 was not overexpressed), the two different tools, our previous approach shows slightly lower bound fractions than Spot-On (Figure 1—figure supplement 3C).

We have now adjusted Abstract, Introduction, Results, and Discussion and made a generally more conservative statement about the role of PBP2 as a limiting factor. We have also included all expression levels and fit parameters into two tables that allow potentially easier comparison between the different fusions, induction levels, and perturbations (Supplementary file 1A,B).

Major changes that must be made to the manuscript before it can be accepted:1) The data are rigorously collected, assembled and analysed, and the findings are in general appropriately discussed in the context of the data obtained and of the existing literature. However, the paper is currently not written for a sufficiently broad audience. The text is too long, not very well structured and sometimes not an easy read. The figures could be more concise and require the reader to continuously go back and forward. Rewriting in a more concise manner (without sacrificing information), would significantly streamline the paper and convey the logic and connections more efficiently. Ways to clarify confusing presentation of the data are further proposed below.

We have now shortened the manuscript at different locations:

1) We have condensed the Introduction, removing a paragraph on the role of MreB filaments for cell-wall integrity and cell shape.

2) We have merged Figures 3 and 6 into a single Figure 3.

3) We have shortened the paragraph on the potential role of Rod-complex components other than MreB for PBP2 binding. Our changes of this paragraph are described in further detail below.

4) We have removed the paragraph about the potential replacement of RodA by other transglycosylases in the Discussion.

2) With the present data, the authors cannot strongly conclude that the formation of the bound PBP2 state is independent of Rod system components other than MreB. […] Also, Figure 3 and 6 might be somehow combined for them to be compared easily, the reader needs to refer to each back and forth (and the multiple supplementary figures) to follow the text.

We agree with the reviewers that our statements on the requirement of Rod-complex components other than MreB were too strong. Accordingly, we have weakened statements in Abstract, Introduction, Results, and Discussion sections. We have now stated more clearly what is supported by our experiments and what not. Finally, we have conducted additional repression experiments, which support our main conclusions as detailed in the following. We now write:

a) RodA: RodA is likely not the binding substrate, as levels are likely repressed at least 2-fold below levels of bound PBP2 molecules, as now also explained in a new Materials and methods section “Calculation of the stoichiometry between RodA and bound

PBP2 molecules”. However, each persistently moving PBP2 molecule (a subfraction of the bound molecules) could still each be associated with a RodA polymerase.

b) MreCD: MreC or MreD could be required for PBP2 binding. However, neither of these proteins is likely the sole, rate-limiting substrate for PBP2 binding. Otherwise we would have expected that diffusive molecules bind at a lower rate upon the 12-fold reduction of MreCD levels observed during the depletion experiment – even if final levels remain higher than PBP2 levels. Constancy of the bound fraction would then require an equal relative change of the unbinding rate, which seems unlikely.

c) RodZ: RodZ might be required for binding and potentially modulates the rate of binding or unbinding directly, as it RodZ depletion reduces the bound fraction by about 3-fold.

We confirmed this effect by an additional repression experiment, using a wildtype background with inducible dCas9 targeting native *rodZ*

**Author response image 2. respfig2:** Cell length, cell width, bound fractions and average diffusion constants as a function of time during RodZ repression in cells carrying inducible dCas9 targeting native rodZ.

The observed effect is very similar to the inducible RodZ protein. However, the phenotype is observed earlier. Specifically, cell width starts to increase already within 2h of depletion. Since we were not able to quantify RodZ levels due to temporal constraints of our proteomics facility, we didn’t include the data in the manuscript.

Despite the strong effect on the bound fraction, we still present evidence that RodZ is likely not the sole, rate-limiting binding substrate for PBP2. We write in the manuscript:

“First, the bound fraction increased after 6 h of depletion, despite a continued low level of RodZ (Figure 5—figure supplement 2F). Second, the spatial pattern of RodZ but not of PBP2 depends on the presence of MreB filaments (Figure 3), suggesting that the localization of PBP2 does also not depend on RodZ. Third, correlations between spatial patterns of mCherry-PBP2 and RodZ-GFP in cells expressing both fusions as sole copies of the respective proteins were low, in contrast to RodZ-GFP and MreB-mCherry (Figure 5—figure supplement 3) (Alyahya et al., 2009; Morgenstein et al., 2015).”

The analysis of spots on the cell periphery and of colocalization between RodZ and PBP2 is now also complemented by qualitatively same experiments in TIRF in an updated Figure 5—figure supplement 3.

3) The explanation given to the FRAP experiment of A22-untreated cells is not satisfying. The authors see no recovery of the bound fraction and indicate that this is presumably because bound molecules left the field of view through persistent motion. It is not clear why this would prevent diffusive molecules to transit to the bound state (immobile included) if the binding sites are not saturated and the average lifetime of a bound molecule is about 1 min.

We have now included a more detailed explanation in the Materials and methods part of why we think that the bound fraction does not recover in the absence of A22 (Materials and methods section “Estimation of the timescale of molecules leaving the TIR field

of view due to persistent motion”). In brief: Molecules that bind, whether initially immobile or persistently moving, transition between immobile and persistent states according to the experimentally measured rates *k*_ip_ and *k*_pi_, respectively. Therefore, molecules binding in a random location in the field of view leave the field of view on average within 45 s. This time is much shorter than the typical time of binding from the diffusive state (calculated as the inverse rate of binding). Therefore, newly binding molecules rapidly leave the field of view due to persistent motion and prevent the bound fraction from recovery.

In the manuscript we write: “The same experiment in non-treated cells did not reveal recovery of the bound fraction (Figure 4—figure supplement 1), likely because bound molecules leave the field of view through persistent motion within less than 1 min (Materials and methods section “Estimation of the timescale of molecules leaving the TIR field of view due to persistent motion”).”

4) Subsection “PBP2 determines the location of new Rod complexes”: In addition to the two possibilities considered by the authors, there is potentially a third possibility: an MreB filament could be present at the PBP2-binding site prior to PBP2 but this filament would not be required for PBP2 recruitment. This could happen if PBP2 and MreB get there independently of each other, recruited by, for ex a third protein. In absence of data showing that PBP2 determines MreB localization this cannot be completely excluded.

We agree with the reviewers the same substrate could recruit both MreB filaments and PBP2 independently, and we have modified the corresponding paragraph in the manuscript (now titled “PBP2 or an unknown low-abundant substrate determines the location of new Rod complexes”).

[Editors' note: further revisions were suggested prior to acceptance, as described below.]

1) No evidence is provided to make the claim that the cell wall itself is the partner that PBP2 is recognizing outside the Rod complex, other than the slow diffusion coefficient of PBP2. The authors should either give more direct evidence that supports this or else rephrase their sentences to clearly indicate that this is speculation. The authors show that MreB is not the binding substrate of PBP2, and that none of the other Rod components provides the sole substrate for PBP2 binding… but no data are provided to directly support that a local cell wall feature binds PBP2 and initiates Rod complex formation. The authors should tone down this conclusion in the Abstract and as a main conclusion from the results reported.

We agree with the reviewer and editors and have now changed the different parts accordingly. In the Abstract we now write “We speculate that the local cell-wall architecture provides the cue for Rod-complex initiation […]”, and we placed this sentence at the end of the Abstract. In the last paragraph of the Introduction we write “We speculate that the cell wall itself provides the cue for Rod-complex initiation.” Finally, in the Discussion paragraph titled “PBP2 might bind directly to the cell wall” we already emphasized the speculative nature of this proposition. Specifically, we already wrote “Based on our observations we speculated that PBP2 might bind to the cell wall directly.”

2) The arguments given to exclude RodZ as PBP2 recruiting factor are not fully convincing on the basis of the data presented. The fraction of bound PBP2 decreases sharply and then progressively increases again (but remains lower than wt levels) during RodZ depletion (from t8). As pointed out during the first round of review, the authors do not show if depletion of RodZ is still operating at later times (the levels of RodZ in the depletion strain are not shown, the text suggests that this is shown in Figure 5—figure supplement 2F but this is not the case). The dCas9 data only shows t3 and t6, while the increase of bound PBP2 fraction operates after t8 in their depletion strain. This point is therefore not fully demonstrated and the conclusion should be toned-down.

We agree with the reviewers that we cannot rule out that RodZ is an important factor for PBP2 recruitment, as already explicitly acknowledged in the Discussion paragraph “What determines the rates of binding and unbinding?”. We now emphasized this possibility by writing in the same paragraph: “However, residual protein levels upon depletion were too high to rule out an important and possibly essential role for PBP2 binding.”

However, we still think that our data provides sufficient evidence to suggest that RodZ is not the sole rate-limiting binding substrate, as stated in the Results and the Discussion. Our CRISPRi-repression data might have caused confusion and we base our argument entirely on the depletion strain and on the localization and colocalization data.

Using the RodZ depletion strain we demonstrate that RodZ levels are stably reduced ~10-fold below WT at 6h and 9h after washing (for mass-spec data see Figure 5—figure supplement 2C), i.e., depletion works until at least 9h (residual expression is likely due to the leaky promoter). Yet, the bound fraction is only reduced by about 40% and 20% w.r.t. WT at the similar time points of t=6h and 8h, respectively. Furthermore, the bound fraction increases with time, while RodZ levels remain stably low. At t=2h, where the reduction of the bound fraction is strongest, RodZ level is expected to be reduced by only 30% w.r.t. WT according to a simple depletion model (*x* = *x*_0_ 2^(-*t/t*_d_), where *x*_0_=1.9 the initial level w.r.t. WT and *t*_d_=81min the doubling time). Therefore, bound fraction and RodZ levels are uncorrelated after 2h of depletion as also summarized in Author response image 3, which depicts the relative change of bound fraction and RodZ levels as a function of time.

**Author response image 3. respfig3:** Relative changes of PBP2 bound fraction and RodZ level w.r.t. WT during RodZ depletion. RodZ levels at 2h and 4h time points are inferred from a simple depletion formula while levels at 0h, 6h, and 9h are from mass spectrometry.

We also still think that our localization experiments provide strong evidence that RodZ is not the sole rate-limiting recruiting factor for PBP2. For any rate-limiting factor we would expect that a strong change of its localization pattern would also affect the localization pattern of PBP2. However, our A22 treatment demonstrates that the localization pattern of PBP2 does not change upon severe change of the localization pattern of RodZ (Figure 3, Figure 3—figure supplement 2). Finally, RodZ and PBP2 localization patterns show very weak correlations (in contrast to RodZ and MreB) (Figure 5—figure supplement 3), further supporting the suggestion that RodZ is not the sole rate-limiting binding factor for PBP2.

3) The explanation that the diffusion constant of the diffusive population of PBP2 decreases significantly upon PBP2 overexpression (“perturbations in the stoichiometry … might lead to alternations of the cell wall structure”) is also pure speculation.

We included this statement in response to the first round of reviewer comments. Without further investigation, we don’t have any other explanation to rationalize this observation. To deemphasize the speculation we now write “Interestingly, overexpression of PBP2 also led to a reduction of the diffusion constant (Figure 1—figure supplement 2). To determine whether this reduction is due to an alteration of the cell-wall structure or due to a different cause, will require further investigation.”

4) Title: please correct "Tranpeptidase PBP2 governs initial localization and activity of major cell-wall synthesis machinery in *Escherichia coli*" to "THE transpeptidase PBP2 governs initial localization and activity of THE major cell-wall synthesis machinery in *Escherichia coli*".

We changed the title accordingly.